# Housing and Food Production: Resident and Grower Perceptions of Peri-Urban Food-Production Landscapes

Shannon Davis *, Guanyu Chen ⬤ and Naomi Darvill

Centre of Excellence: Designing Future Productive Landscapes, School of Landscape Architecture,
Lincoln University, Lincoln 7647, New Zealand; hanley.chen@lincolnuni.ac.nz (G.C.);
naomi.darvill@lincolnuni.ac.nz (N.D.)
* Correspondence: shannon.davis@lincoln.ac.nz; Tel.: +64-21-121-1298

**Abstract:** The loss of productive soils and food-producing landscapes on the edges of cities is an increasing issue facing Aotearoa New Zealand. Like many countries globally, New Zealand's largest cities are facing rapid expansion because of increasing urbanisation, with high levels of low-density residential sprawl into the productive peri-urban hinterlands and increasing rates of 'reverse sensitivity'. Food production, as a result, is being pushed further away, disconnected from the communities it serves, and often onto less productive soil. This paper explores the perceptions and attitudes of both peri-urban residents and food producers living and working within the peri-urban zone of Ōtautahi Christchurch. Conducting two surveys, one with residents and another with producers, respondents' perceptions of food growing within this peri-urban landscape are explored to better understand the enablers and barriers of growing food close to cities. Overall, the results indicated that peri-urban residents appreciate food being produced close to where they live, with over 90% of residential respondents feeling either 'mostly positive' or 'extremely positive' towards food being grown close to their homes. Of greatest concern for peri-urban residents were issues relating to negative impacts on the environment and human health, with particular concern for water quality. The lack of accessibility to locally produced food was also identified as an area of concern to residents. Food producers felt less positive towards operating their food-production enterprises within the peri-urban zone, identifying a range of issues impacting their experience. The information rendered from these surveys provides a base for future land-use planning consideration within the peri-urban zone, where both food production and housing can co-exist.

**Keywords:** peri-urban; food-production landscapes; highly productive soil; residential expansion; reverse sensitivity; community perception; grower/farmer perception

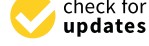



## 1. Introduction

Historically, cities and urban settlements were located in well-connected landscapes with good access to natural resources such as fresh water and fertile soil. These landscapes supported the production and distribution of nutritious and diverse food. Population growth and rapid urbanisation, however, pose complex issues for urban food sustainability, and, critical to this, is access to existing and potential land for agriculture at the fringes of cities [1–7].

As cities grow, the loss of productive land on their edges, and therefore the ability to contribute to urban food security, is a topic of increasing concern for cities of the Global North. Urban and peri-urban agriculture are now considered critical components to providing accessible food to maintain a healthy urban population [1,8–10]. Sarker et al. [11] propose that there is a strong consensus among policymakers that urban and peri-urban agriculture should be a vital part of planning processes and urban design. However, in recent decades, planning has not only overlooked food systems as a vital and necessary component of city planning, but in much of the Global North it has actively contributed

to its movement away from urban settlements [12]. Opitz [9] describes how while urban agriculture can meet the food needs at a household level, peri-urban food production can provide larger quantities with broader distribution pathways, therefore contributing to overall urban food security.

The loss of priority for urban and peri-urban agriculture in land-use planning is a global trend [13], with cities often favouring rural transformation [14]. In recent decades, however, urban food has become a nexus of increased public attention focused on the dislocation caused by globalized food systems and the resulting impacts on environmental sustainability and urban resilience. Well-functioning peri-urban food systems are today understood as an important part of enabling urban food resilience [9,11,15,16].

The decoupling of cities from sources of food supply is a unique exception to a long history of urban–rural relations [13]. For Aotearoa New Zealand, approximately 15% of the land is categorised as 'highly productive' [17], and much of this sits at the edges of the main cities. Since 2002, 35,000 hectares of land classed as 'highly productive' has been lost to urban development [17]. The biggest losses have been seen in the areas of Tāmaki Makaurau Auckland and Ōtautahi Christchurch, and were primarily due to urban expansion.

Aotearoa New Zealand's peri-urban landscapes are today facing unprecedented pressure for land-use change due to residential sprawl. In addition to the physical pressure of land-use change are issues relating to land fragmentation and 'reverse sensitivity' [18]. The term 'reverse sensitivity' describes the vulnerability of an established type of land use, such as farming, to legal complaints from a newly established land use, such as residential development [19]. In response to the ongoing pressure and loss of land for food production, the Aotearoa New Zealand Ministry for the Environment gazetted, in 2022, the National Policy Statement on Highly Productive Land (NPS-HPL). This piece of national-level legislation aims to protect and retain the remaining land classed as 'highly productive' from land-use change.

Protecting and retaining land within the peri-urban zone for food production provides multiple benefits for the environment and society, as well as the economic well-being of farmers and the communities they are connected with [20]. As a unique form of productive landscape in its spatial proximity to urban centres, the peri-urban zone delivers dynamic and multi-functional benefits to cities through the provision of essential ecosystem services such as flood mitigation, urban cooling, amenities and food production. Langemeyer [13] argues that urban sustainability, resilience and multi-functionality are all increased with urban and peri-urban agriculture, and that it *"deserves a much stronger consideration in planning for urban resilience and global sustainability strategies"* (p. 2).

The peri-urban zone of Ōtautahi Christchurch offers an interesting area for analysis as it is not only experiencing a period of significant urban growth, but also has a strong agricultural history [21]. Agriculture remains the dominant land use, with diverse farming production types ranging from livestock rearing to crop raising [22]; however, the zone is experiencing significant expansion of residential developments. This study surveyed residents living within Waikirikiri Selwyn District, a district within the Ōtautahi Christchurch peri-urban zone (Figure 1).

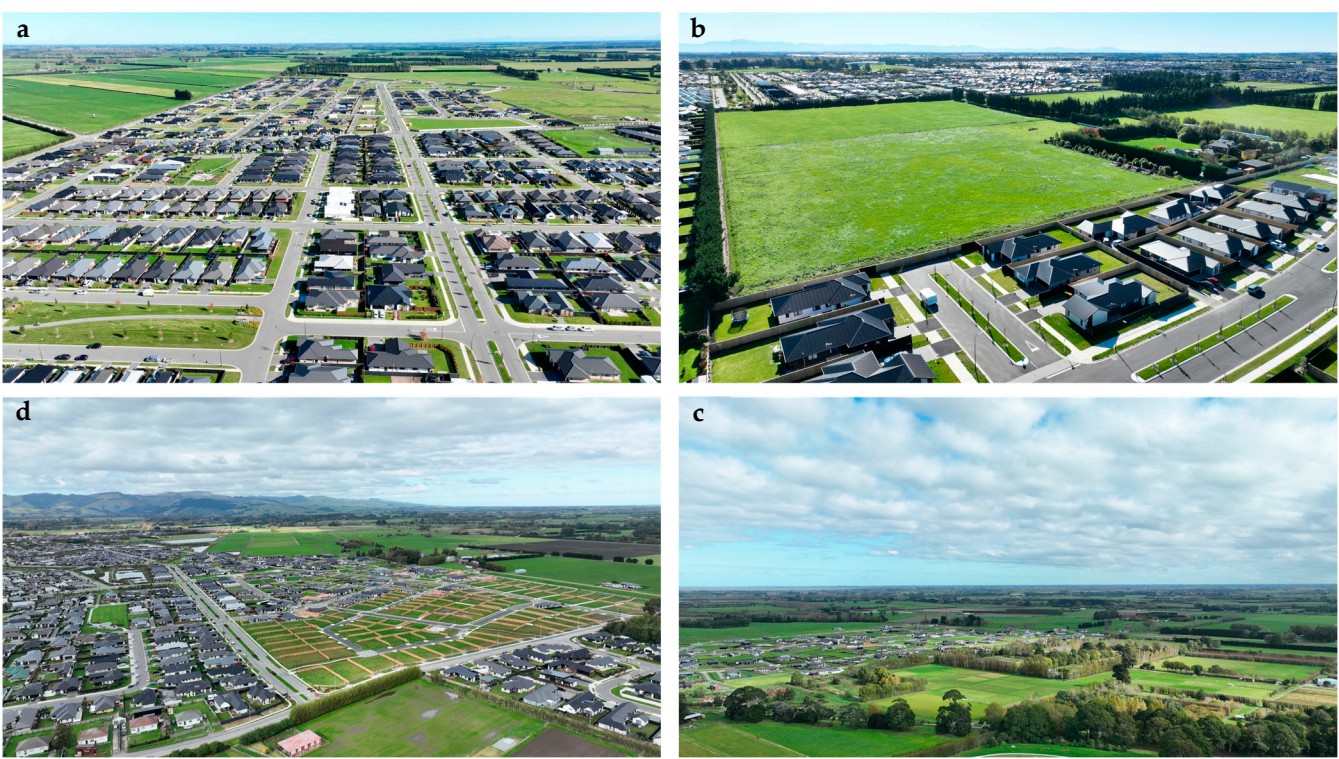

**Figure 1.** Residential sprawl onto productive land within the Ōtautahi Christchurch peri-urban zone. Clockwise from top left: (**a**) new greenfield low-density housing sprawl in Rolleston, (**b**) a farm in Rolleston surrounded by a new housing subdivision with roads 'waiting' to be connected through the farm, (**c**) recent 'rural residential' and 'lifestyle block' subdivision on highly productive soils in Lincoln, (**d**) sprawl continuing in Lincoln with another phase of urban development getting underway (Image credit: Donald Royds 2023).

## 2. Methods

The method adopted for this study encompassed two surveys. The first survey involved residents living in three towns in the Waikirikiri Selwyn District. This survey sought to understand residents' perceptions toward food-production landscapes close to their homes, as well as resident experiences with different production types and farming approaches. This survey also investigated the potential of household-based food production (food grown by participants at their own homes). The second, smaller survey, sought to understand the perceptions of food producers (growers and farmers) operating within the same zone. By surveying both groups a holistic understanding of the overall enablers and barriers to retaining peri-urban landscapes for productive means was pursued.

### 2.1. Survey of Residents

#### 2.1.1. Study Area

Residents of Lincoln, Rolleston and Darfield townships (Figure 2) within the Waikirikiri Selwyn District were surveyed. Over the past 25 years, this district has seen significant population growth, from around 27,600 residents in the year 2000 to over 79,000 in 2022 (an increase of 186%) [23]. In 2021, the district grew by 4.9%, which was the second fastest district growth in the country. The district has also seen a rapid increase in residential development activity in terms of dwelling construction, which has consistently exceeded projections [23]. This rapid growth is projected to continue into the foreseeable future.

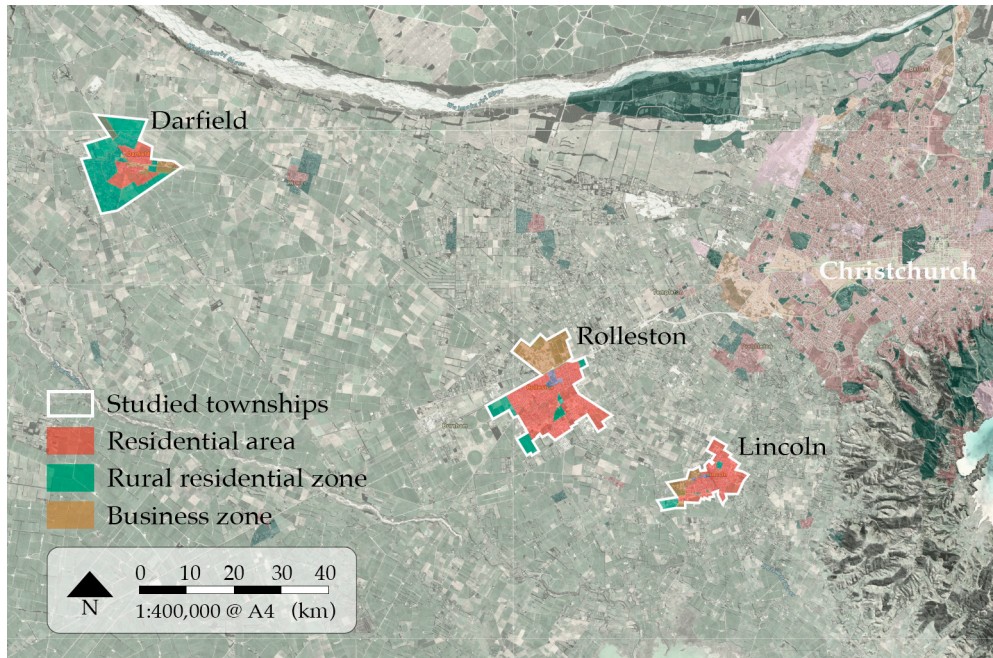

**Figure 2.** Resident survey study areas of Lincoln, Rolleston and Darfield, and their spatial relationship to Ōtautahi Christchurch city (Adapted from Land Information New Zealand, licensed under Creative Commons Attribution 4.0 International (CC BY 4.0) [23]).

The survey of food producers recruited participants from a slightly wider geographical area, extending into the Waimakiriri District, which forms part of the peri-urban zone to the north of Ōtautahi Christchurch.

Lincoln is the second largest town within Waikirikiri Selwyn District and is approximately 20 km south of Christchurch city. It has traditionally been a small rural service town primarily developed around the agricultural sector, Lincoln University (New Zealand's land-based university) and several Crown Research Institutes. The township is located on the highest class soil categories of LUC 1–3, defined as 'highly productive' (Figure 3). The township has experienced significant population growth and substantial urban extension in recent years, growing from approximately 2500 people in 2000 to approximately 9000 in 2022 (a 260% increase) [24]. Residents have been vocal in their concerns about residential growth encroaching on the highly productive soil that encompasses the town. They have also voiced concern over the impact of land-use change that threatens to alter the town's character and identity as a 'rural service town'.

Rolleston is the largest township within the Ōtautahi Christchurch city commuter belt. It is located 22 km southwest of the city and has a rapidly growing population, including the stimulus of the rapid movement of 'urban' residents to the peri-urban township after the 2010/2011 Canterbury earthquake series. The township has grown from about 3000 residents in the early 2000s to over 25,000 residents in 2022 (a 730% increase) [25]. Rolleston is also surrounded by significant areas of 'highly productive soil'; however, this is to a lesser extent than Lincoln (Figure 3).

Darfield is located approximately 50 km from the city. It is primarily a rural service town, although recent growth in residential subdivision has seen a greater commuter population located in the town. Like Lincoln, Darfield has a strong history based on agricultural land use, which continues today. With a population of approximately 3000 residents [26], Darfield was the smallest township in the study. It is also located in areas of highly productive soils (Figure 3).

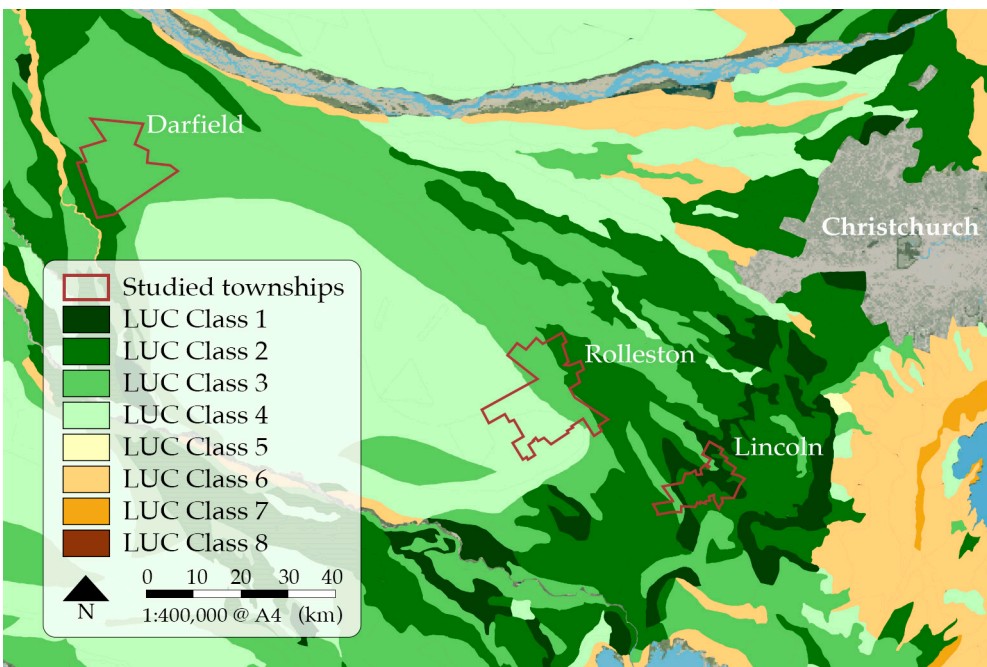

**Figure 3.** Land-Use Capability (LUC) map illustrating the soil productivity of the land around the townships of Lincoln, Rolleston and Darfield. This soil classification system categorises land into eight classes according to the physical qualities of the land, soil and environment. LUC classes 1–3 are considered 'highly productive'. LUC Class 1 is categorised as arable and is the most versatile multiple-use land, with minimal limitations, which is highly suitable for cropping, viticulture, berry fruit, pastoralism, tree crops and forestry. LUC Class 2 is also categorised as arable with very good multiple-use land, with slight limitations, but suitable for cropping, viticulture, berry fruit, pastoralism, tree crops and forestry. LUC Class 3 is also considered arable with moderate limitations restricting crop types and intensity of cultivation, but suitable for cropping, viticulture, berry fruit, pastoralism, tree crops and forestry. LUC Class 4 is considered arable but with significant limitations for arable use or cultivation, very limited crop types, and suitable for occasional cropping, pastoralism, tree crops and forestry. Some Class 4 land is also suitable for viticulture and berry fruit. LUC Class 5 is considered non-arable but is highly productive pastoral land, not suitable for crops but with only slight limitations to pastoral, viticulture, tree crops and forestry. LUC Class 6 is also considered non-arable, having slight-to-moderate limitations to pastoral use, being suitable for pasture, tree crops, forestry and, in some cases, vineyards. LUC Class 7 is described as non-arable with moderate-to-very severe limitations to pastoral use, having a high risk of land requiring active management to achieve sustainable production. This classification can be suited to grazing with intensive soil conservation measures but is more suited to forestry. Finally, LUC Class 8 is non-arable land with very severe-to-extreme limitations to all productive land uses. (Adapted from Our Environment by Manaaki Whenua Landcare Research, licensed under Creative Commons Attribution 4.0 International (CC BY 4.0) [27]).

As Aotearoa New Zealand responds to what has been termed a 'national housing crisis', councils around the country have been tasked with testing the capacity of their districts to provide for the housing needs of their communities. Both Lincoln and Rolleston are included in the Selwyn District Council's 'medium-density residential strategy', with Darfield identified as a 'future growth area' [27]. This indicates that significant residential development in each township will continue, putting further pressure on land-use change, and potential further growth in land-use conflict and 'reverse sensitivity' issues.

This study therefore seeks to understand the perceptions and attitudes of the peri-urban community (residents and food producers) towards food production in the peri-urban zone. Through this research, the enablers and barriers to retaining food production within this zone into the future were identified.

2.1.2. Sampling and Surveying Procedure

For the survey of residents, a total of 2060 questionnaires were distributed within the three townships. This ensured enough respondents were sampled to statistically represent the 37,420 residents. Of the 2060 questionnaires, 1423 were distributed to Rolleston, 430 to Lincoln and 207 to Darfield, which covers a similar percentage (approximately 5.5%) of the population in each town.

A transect approach was used to ensure the samples represented the spatial distribution of the studied population. As shown in Figure 4, a transect wedge covering approximately 5.5% of each township was drawn to represent the population's spatial distribution from the town centre to the urban–rural interface. The sampled areas were then divided into three zones according to their distance to the town centre. The areas within one third of the radial transect from the town centre were defined as Zone 1 (shown in red in Figure 4) covering central township residents. The peripheral one third of the transect was defined as Zone 3 (shown in blue), covering the residents living closest to the rural interface. The remaining third, sitting between Zone 1 and Zone 3, was defined as Zone 2 (shown in green), covering the residents in between. Each of the coloured blocks in Figure 4 represents one residential household that was sampled for the survey.

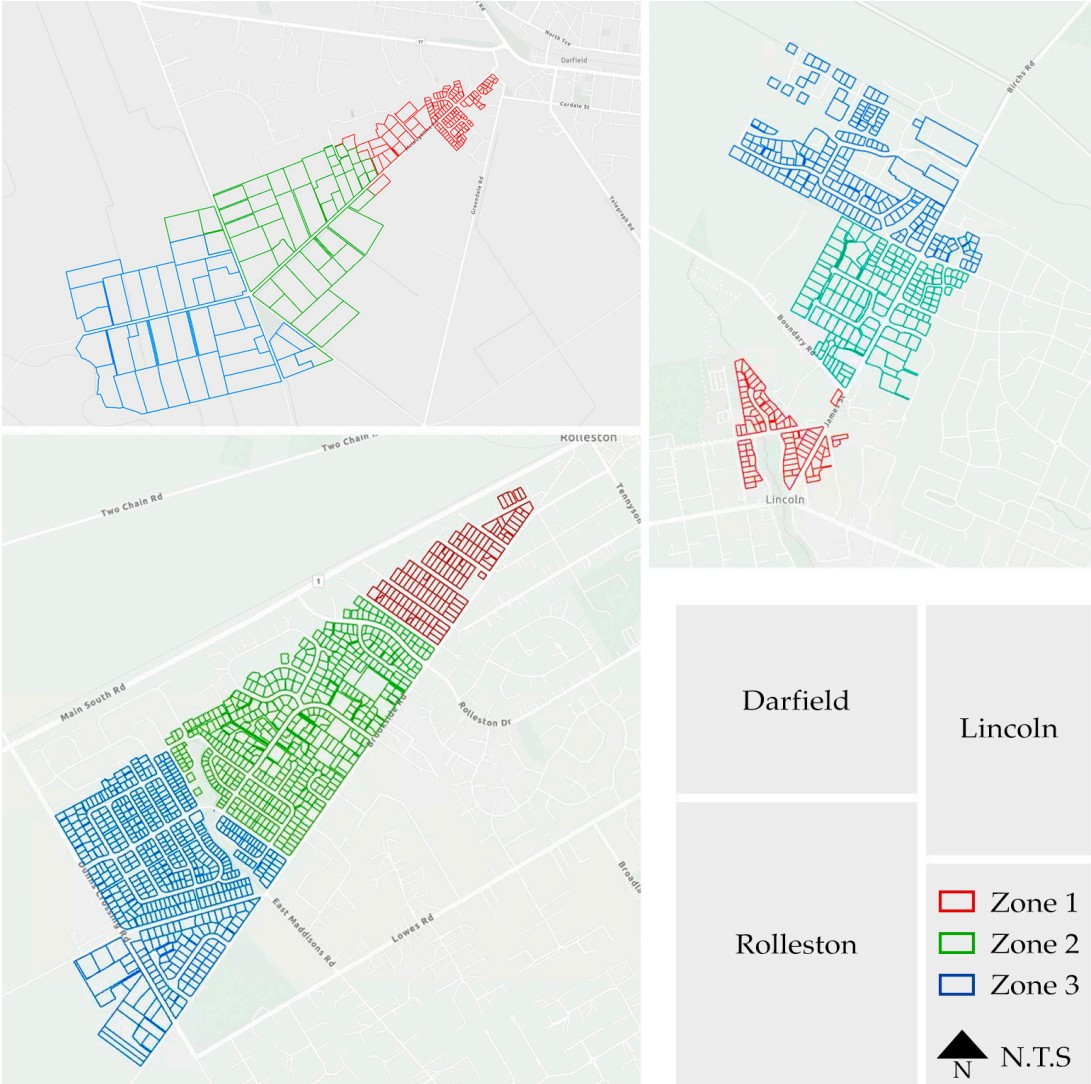

**Figure 4.** Survey distribution transect for Lincoln, Rolleston and Darfield (adapted from Land Information New Zealand, licensed under Creative Commons Attribution 4.0 International (CC BY 4.0)) [6].

Surveys were distributed to the selected households via a letterbox drop. Criteria for leaving a survey included being a residential property located within one of the study areas and with the ability to leave and then collect a completed survey from a letterbox. Participants were directed that one resident per household over the age of 18 was able to participate.

Human ethics approval to conduct the surveys within this study was granted by the Lincoln University Research Management Office, Human Ethics Committee, Application No: 2022-42.

2.1.3. Questionnaire Design—Resident Survey

The questionnaire consisted of 20 questions, including multiple-choice questions, Likert-scale questions, matrix questions, and open-ended questions. At the beginning of the questionnaire, respondents were asked to answer five sociodemographic questions, including their age bracket, gender, household type (whom they were living with), property ownership (whether they own or rent their property) and their level of connection to rural farming landscapes (shown in Table 1). Next, respondents were asked five questions about the characteristics of their property. The questions included how long they have lived at the property, the approximate age of the house, whether the property directly borders land that commercially produces food, the land unit type (either residential section, lifestyle block (see note 2) or farm), and finally, the approximate land size of the property. The data collected through these sociodemographic and property-related questions were considered as explanatory variables to explain and predict the outcome variables collected through the latter questions in the survey.

**Table 1.** Questionnaire design.

|  | Variables | Measurement Scale |
|---|---|---|
| Sociodemographic | Age | Ordinal |
|  | Gender | Binary |
|  | Household type | Categorical |
|  | Property ownership | Binary |
|  | Farming background | Binary |
| Property characteristics | Town | Categorical |
|  | Zone | Ordinal |
|  | Number of years lived in the surveyed property | Ordinal |
|  | Age of the surveyed property | Ordinal |
|  | Direct property–commercial farm interface | Binary |
|  | Property type | Categorical |
|  | Property land size | Ordinal |
| Behaviour and awareness | Type of home produce | Binary |
|  | Usage of home produce | Binary |
|  | Personal connection to the food producers | Ordinal |
|  | Known channels for purchasing local produce | Binary |
| Perception | Overall perception towards food-producing landscapes | Likert scale |
|  | Perception towards specific types of production | Likert scale |
|  | Perception towards specific food-producing/gathering approach | Likert scale |

A range of outcome variables were collected to explore the respondents' food production-related behaviour, awareness and perception/attitude. As shown in Table 1, the respondents were asked whether they produce any food at home, and if so, what they produce. If they did produce food at home, the following question asked them what they do with the food they produce. After the two behaviour-related questions, the respondents were asked whether they were aware of any channels of purchasing local produce and what channels

they are (e.g., farm-gate sales or farmers' markets). They were also asked whether they personally know the farmers who produce the food they purchase locally.

Next, a 5-point Likert scale (extremely positive, mostly positive, neutral, mostly negative and extremely negative) was used to assess respondents' overall attitude towards the food-producing landscapes that are located close to where they live. To achieve a deeper understanding of the factors influencing residents' feelings towards food-producing landscapes, the questionnaire also measured the respondents' attitudes towards specific types of production and a range of food-producing approaches, by using the same Likert scale in two matrix questions. The respondents were also given an opportunity to explain the reasons why they like or do not like the types of production practices they rated by providing comments after the matrix question.

At the end of the questionnaire, respondents were asked to answer two open-ended questions focused on the reasons why they liked or did not like having food-production landscapes close to their homes.

### 2.1.4. Data Analysis

The quantitative data collected through the 'tick-box' questions and the qualitative data collected through the open-ended questions were analysed separately.

The quantitative data were analysed descriptively and examined by testing the significance of inter-group differences and correlations between the outcome and predictor variables using SPSS 28. A range of statistical tests and models were employed, including an ordinal logistic regression model, Spearman's rank correlation test, Mann–Whitney U tests, and Kruskal–Wallis tests, depending on the variable types (binary, categorical, ordinal or scale). An ordinal logistic regression model was built to examine the relationships between the residents' overall perceptions of food-producing landscapes and all the predictor variables. Spearman's rank correlation tests were conducted to examine the relationship between the outcome variable and the ordinal predictor variables, such as the respondents' age, the land size of their property, and the age of their house. Mann–Whitney U tests were conducted to examine the relationship between the outcome variable and the binary predictor variables, such as the respondents' gender, whether they have a farming background, and whether they produce food at home. Kruskal–Wallis tests were conducted to examine the relationship between the outcome variable and the categorical predictor variables, such as the type of land title (residential, lifestyle block or farm).

The qualitative text data (i.e., participants' written responses to the two open-ended questions, (1) What do you like about living near a food producing landscape? and (2) What do you not like about living near a food producing landscape?) were first transcribed and then coded using a line-by-line coding approach. This coding approach draws all the identifiable opinions or ideas indicated by the respondents from their text responses and then structures the codes according to the inter-relationship between them to extract the shared or divergent opinions of the surveyed population.

### 2.2. Food Producer Survey

The food producer survey was smaller, eliciting nine completed questionnaires that had been distributed to stall holders from local farmers' markets that host growers and farmers who operate their enterprises within the peri-urban zone of Ōtautahi Christchurch.

### Questionnaire Design—Food Producer Survey

The data analysis approach to the food producer survey was the same as for the residents' survey, as explained in Section 2.1.4. However, the results from this survey were not tested for inter-group differences or correlations, as the sample size was relatively small and did not meet some of the assumptions of the statistical tests and models employed in the analysis of the resident survey.

## 3. Results

### 3.1. Resident Survey—Response Rate

As shown in Table 2, the survey yielded 190 responses in total, of which 179 were accepted and 11 were rejected due to incomplete consent forms. The overall response rate was 8.69%. According to the latest census data available at the time of conducting the survey, there was a total population of 37,420 (see note 1) residents residing within the three surveyed towns. Yielding 179 valid responses, therefore, meant that the sampling statistically represented the target population with a 7% margin of error at a 95% level of confidence.

**Table 2.** Resident survey distribution and response rate by township and zone.

|  | Rolleston | | | Lincoln | | | Darfield | | | Total |
| --- | --- | --- | --- | --- | --- | --- | --- | --- | --- | --- |
|  | Zone 1 | Zone 2 | Zone 3 | Zone 1 | Zone 2 | Zone 3 | Zone 1 | Zone 2 | Zone 3 | Total |
| Distributed | 180 | 589 | 654 | 79 | 136 | 215 | 126 | 47 | 34 | 2060 |
| Response collected | 15 | 34 | 36 | 21 | 30 | 18 | 19 | 11 | 6 | 190 |
| Response accepted | 15 | 31 | 34 | 20 | 25 | 18 | 19 | 11 | 6 | 179 |
| Response rate | 8.33% | 5.26% | 5.20% | 25.32% | 18.38% | 8.37% | 15.08% | 23.40% | 17.65% | 8.69% |

### 3.2. Resident Survey—Demographic Characteristics

Table 3 outlines the demographic profile of the respondents. Overall, the respondent population was relatively gender-balanced, mostly living with family members, with most respondents owning the surveyed property (as opposed to renting it). People aged 65 or over made up more than 40% of the respondents, and almost one third of the respondents were aged 50–64. By contrast, younger respondents aged 18–29 and 30–39 accounted for just 5% and 14% of the total, respectively. Over one third of the survey's respondents had no former background in farming.

**Table 3.** Sociodemographic profile of respondents (all percentages shown below are relative to the number of respondents who provided a valid answer to that question).

| Attribute | Description | Count | N | Percentage |
| --- | --- | --- | --- | --- |
| Age bracket | 18-29 years old | 8 | | 4.52% |
| | 30-39 years old | 25 | | 14.12% |
| | 40-49 years old | 19 | 177 | 10.73% |
| | 50-64 years old | 50 | | 28.25% |
| | 65+ years old | 75 | | 42.37% |
| Gender | Female | 97 | | 54.49% |
| | Male | 81 | 178 | 45.51% |
| | Prefer to self describe | 0 | | 0.00% |
| Household type | One person | 29 | | 16.20% |
| | Couple | 76 | | 42.46% |
| | House share | 4 | 179 | 2.23% |
| | Family (incl. child/children) | 63 | | 35.20% |
| | Family (excl. child/children) | 7 | | 3.91% |
| Property ownership | Own | 170 | 178 | 95.51% |
| | Rent | 8 | | 4.49% |
| Farming background | Grew up on a farm | 39 | | 21.91% |
| | Have lived in a farming community | 53 | | 29.78% |
| | Have worked on a farm | 25 | 178 | 14.04% |
| | Regularly visit(ed) a farm (or farms) | 19 | | 10.67% |
| | Others | 10 | | 5.62% |
| | No former connection with farming | 60 | | 33.71% |

### 3.3. Resident Survey—Property Characteristics

As shown in Table 4, more than half of the residents surveyed had moved into the surveyed properties during the seven years 2015–2021, which echoes the rapid expansion of residential development within the Waikirikiri Selwyn District townships discussed above. Prior to that, approximately 20% of respondents moved into their current property within the period 2009–2015. The age of the surveyed property also reflected a similar trend. More than a quarter of the surveyed properties were built between 2015 and 2018 (Figure 5 illustrates the housing typology and form of recent residential developments in the district). The number of properties built within that 3-year period was twice the average number within other three-year periods before 2015 or after 2018. Regarding the 'type of property', residential sections were the majority, in comparison to lifestyle blocks (see note 2) and farms. More than 90% of surveyed properties did not directly border a commercial farm. Around half of the properties were sized from 500 to 1000 square meters, while one fifth of the properties were smaller than 500 square meters, and one third were larger than 1000 square meters.

**Table 4.** Property profile of respondents (all percentages shown below are relative to the number of respondents who provided a valid answer to that question).

| Attribute | Description | Count | N | Percentage |
|---|---|---|---|---|
| Number of years lived in the surveyed property | Less than one year | 10 | | 5.68% |
| | 1-4 year | 47 | | 26.70% |
| | 4-7 year | 52 | 176 | 29.55% |
| | 7-10 year | 18 | | 10.23% |
| | 10-13 year | 18 | | 10.23% |
| | 13 years or more | 31 | | 17.61% |
| Age of the surveyed property | Less than one year | 1 | | 0.56% |
| | 1-4 year | 23 | | 12.99% |
| | 4-7 year | 45 | 177 | 25.42% |
| | 7-10 year | 20 | | 11.30% |
| | 10-13 year | 23 | | 12.99% |
| | 13 years or more | 65 | | 36.72% |
| Direct property - commercial farm interface | Yes | 14 | 176 | 7.95% |
| | No | 162 | | 92.05% |
| Property type | Section | 160 | | 91.43% |
| | Lifestyle block | 13 | 175 | 7.43% |
| | Farm | 2 | | 1.14% |
| Property land size | Less than 500 sqm | 33 | | 19.19% |
| | 500-1000 sqm | 84 | 172 | 48.84% |
| | 1000-3000 sqm | 33 | | 19.19% |
| | 3000+ sqm | 22 | | 12.79% |

### 3.4. Resident Survey—Behavioural Characteristics and Awareness

Table 5 illustrates the behavioural characteristics and awareness of the respondents towards their local food-producing landscapes. Overall, the results showed that a high percentage (90%) of respondents produce at least one type of produce at home. Vegetables were the most common option, with more than 80% of respondents growing these at home, followed by fruit (52%), herbs (52%) and berries (44%). In comparison to these plant-based options, animal-based production was less common for residential home growers—8% of respondents produced meat products at home, 7% produced eggs and 2% produced honey. The survey revealed that food produced at home was normally consumed by the household,

with any excess being shared with family and friends. There were also 6% of households that sold their produce, and 3% who contributed produce to a community pantry.

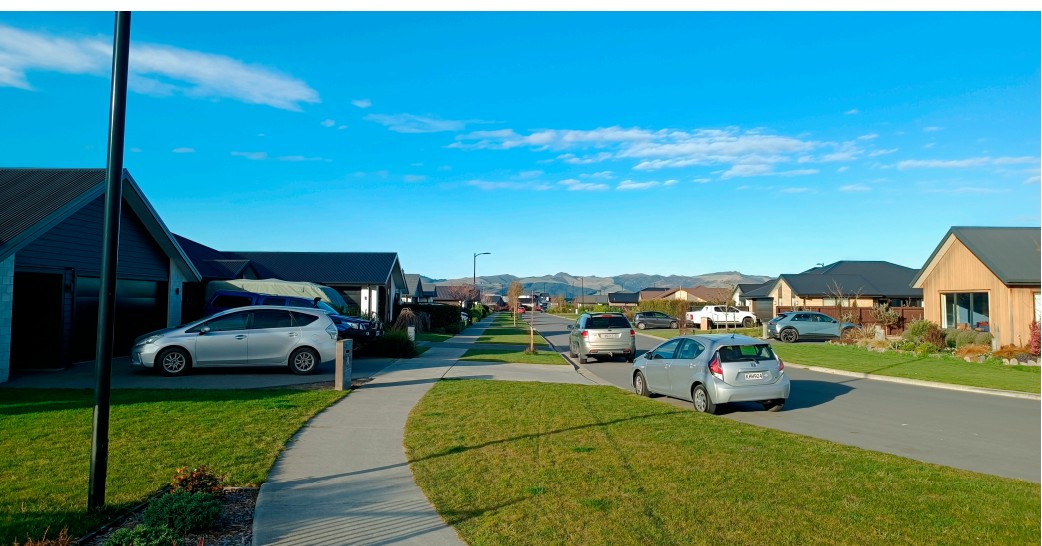

**Figure 5.** Typical house and street typology of new greenfield residential subdivision development within the Selwyn District (Author, 2023).

**Table 5.** Behavioural profile of respondents (all percentages shown below are relative to the number of respondents who provided a valid answer to that question).

| Attribute | Description | Count | N | Percentage |
|---|---|---|---|---|
| Type of home produce | Fruit | 93 | | 52.25% |
| | Berries | 79 | | 44.38% |
| | Vegetables | 143 | | 80.34% |
| | Eggs | 13 | 178 | 7.30% |
| | Honey | 4 | | 2.25% |
| | Herbs | 92 | | 51.69% |
| | Meat products | 15 | | 8.43% |
| | Do not produce | 18 | | 10.11% |
| Usage of the home produce | Household consumption | 157 | | 98.13% |
| | Give away to family and friends | 77 | 160 | 48.13% |
| | Sell | 9 | | 5.63% |
| | Community pantry | 4 | | 2.50% |
| Known channels for purchasing local produce | Farm gate | 84 | | 47.19% |
| | Farmers market | 68 | | 38.20% |
| | Local retail stores | 70 | 178 | 39.33% |
| | Other channels | 21 | | 11.80% |
| | No known channels | 44 | | 24.72% |
| Personal connection to the food producers of the local produce I purchased | I know all of them | 2 | | 1.49% |
| | I know some of them | 60 | 134 | 44.78% |
| | I don't know any of them | 72 | | 53.73% |

Table 5 also illustrates that more than three quarters of respondents knew at least one channel of purchasing local produce. Farm-gate sales was the most acknowledged channel, being recognised by 47% of respondents. Approximately 40% of respondents acknowledged local retail stores and farmers' markets (Figure 6 illustrates a typical farmers' market within the district) as channels for purchasing local produce. Among the 134 respondents who knew at least one channel of purchasing local produce, about half of them had a personal connection to those producers.

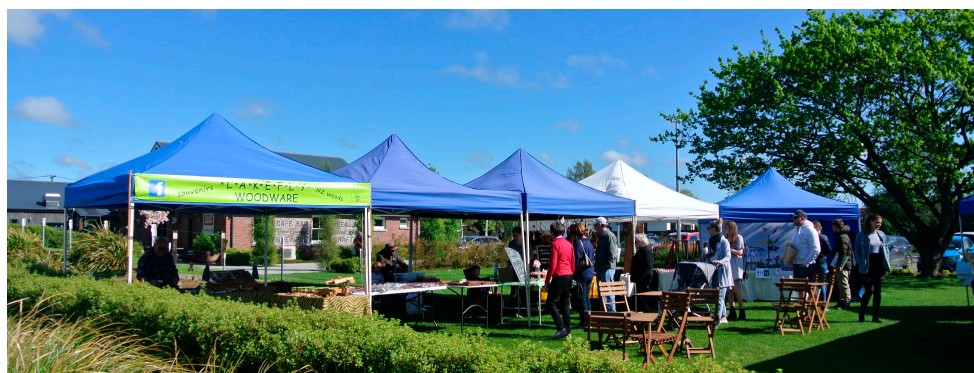

**Figure 6.** The Lincoln Farmers' Market where residents and food producers connect within the neighbourhood setting (Author, 2019).

*3.5. Resident Survey—Perceptions and Attitude*

3.5.1. Overall Perception towards Farming Landscapes

Overall, the survey revealed that residents predominantly feel positively towards the food-producing landscapes that are located close to where they live. As shown in Table 6 and Figure 7, no respondents felt extremely negative, and less than 2% of respondents indicated negative feelings towards food-producing landscapes close to their homes. On the contrary, more than 60% of respondents felt 'extremely positive', and 32% felt 'mostly positive' towards those landscapes.

**Table 6.** Respondents' perception towards food-producing landscapes (all percentages shown below are relative to the 178 respondents who indicated their attitudes toward food-producing landscapes).

| Attribute | Description | Count | N | Percentage |
|---|---|---|---|---|
| Overall perception towards food-producing landscapes | Extremely positive | 107 | | 60.11% |
| | Mostly positive | 57 | | 32.02% |
| | Neutral | 11 | 178 | 6.18% |
| | Mostly negative | 3 | | 1.69% |
| | Extremely negative | 0 | | 0.00% |

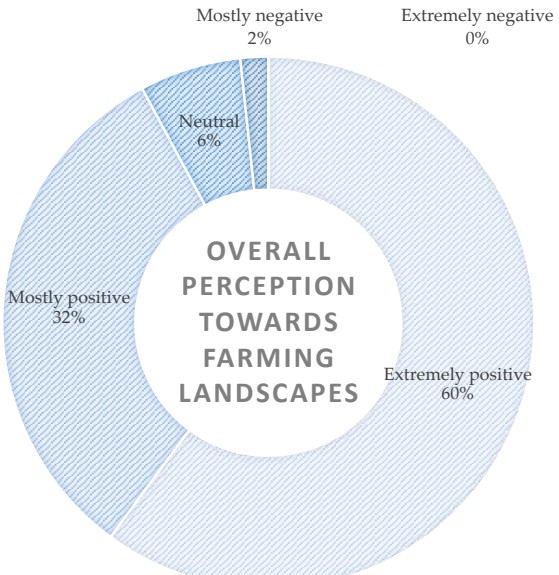

**Figure 7.** Residents' overall perception towards food-producing landscapes (all percentages shown below are relative to the 178 respondents who indicated their attitudes toward food-producing landscapes).

To gain a comprehensive understanding of the reasons why respondents felt positive or negative towards the food-producing landscapes close to where they live, participants were asked to provide a brief explanation of anything they like or dislike about living near these landscapes. Their comments were coded as per the method explained in Section 2.1.4. Figure 8 presents the respondents' stated reasons for their positive perceptions, along with some representative comments. The most frequently cited reasons, mentioned by over one third of respondents, were associated with produce-related benefits, with access to local produce (47, 26%), fresh and healthy produce (34, 19%), and being cost-effective (14, 8%) the most highly cited reasons for this.

Following the produce-related benefits, perceptual benefits offered by the farms (including such aspects as a rural outlook and the perception of being 'green') were indicated by around 30% of respondents. The educational value of farming landscapes emerged as the third most common category, cited by about one fifth of respondents. Additionally, 16% of responses revealed a positive perception towards the farming community 'as a whole' and expressed a desire to support local growers and farmers. Beyond this, Figure 8 also illustrated some less-frequently mentioned reasons, including the associative benefits (23, 13%) and socio-economic benefits (22, 12%) provided by the food-producing landscapes, the fact that the lands are being used productively (15, 8%), the environmental benefits (14, 8%), the respondents' personal connection to farming practices (11, 6%) and the perception that most farms were operated in a responsible way (6, 3%).

Similarly, Figure 9 provides an overview of the respondents' reported reasons for their negative perceptions, accompanied by representative comments. Overall, the reasons for negative perceptions can be categorised into six general groups. The three primary categories include the impacts on human health, on residents' quality of life and on the environment, each of which were mentioned by 26% of respondents. Some specific issues were included in more than one category. For example, chemical spray and water pollution were included in both 'impacts on human health' and 'impacts on the environment'. In addition to these identified issues, chemical spray (21, 12%) water pollution (20, 11%), dust (5, 3%), air pollution (4, 2%), general health issues (3, 2%) and pests or weeds (2, 1%) were also commented upon. Other issues that relate to the environment included excessive water usage (6, 3%), generally stated environmental issues (5, 3%) and impacts on biodiversity (4, 2%).

Issues mentioned that affected residents' quality of life included the undesirable smell from farms (21, 12%), traffic-related issues (9, 5%), noise (6, 3%), insufficient access to local produce (5, 3%), poor experience of buying local produce (3, 2%), water shortage caused by agricultural usage (2, 1%), animal droppings (2, 1%) and pests or weeds (2, 1%). Beyond these three major groups of issues, ethical concerns of farming practices were also mentioned by 6% of respondents.

An ordinal logistic regression model was employed to investigate whether the predictor variables indicated in Tables 3–5 predicted the residents' overall perception towards food-producing landscapes. The model showed a poor fit ($p = 0.906$). A range of adjustments were made to the original model by adjusting the predictor variables entering the model, with just two of the adjusted models showing good model fit. In the first model, the two entered predictors—township and ownership—together accounted for a significant amount of variance in the outcome (likelihood ratio $\chi^2(3) = 8.911$, $p = 0.032$). Only township (township = Lincoln) significantly independently predicted the outcome variable, perception ($B = 0.962$, $SE = 0.363$, $p = 0.008$). In the second model, similarly, township and farming background together accounted for a significant amount of variance in the outcome (likelihood ratio $\chi^2(3) = 8.081$, $p = 0.044$), and only township (township = Lincoln) significantly independently predicted the respondents' perception ($B = 0.866$, $SE = 0.359$, $p = 0.016$).

**Produce-related benefits (68, 38%)**
Access to local produce (47, 26%)
Fresh and healthy produce (34, 19%)
Cost-effectiveness of local produce (14, 8%)
Quality produce (9, 5%)
Convenience (3, 2%)
Enjoy the produce (3, 2%)

"Lots of fresh locally grown food and veges are easily accessible, at both markets and gate sales. Often these are cheaper than supermarket."

"Seeing good land use and healthy, well looked after products."

"Like having fresh food available to purchase keeping me healthy..."

"Buying local would mean better quality and hopefully the produce would be fresher."

"It is amazing to have access to such a wide range of produce and such a convenient and coordinated way."

**Perceptual benefits (55, 31%)**
Farming landscape, rural outlook, and associated lifestyle (47, 26%)
Nice open space or sight (16, 9%)
Being 'green' (6, 3%)
Smells (4, 2%)
Sounds (2, 1%)

"I like the openness of the rural landscape, the feeling of being part of 'traditional' New Zealand country life."

"We came to Lincoln to be rural, have the wide open spaces and fresh air. We much prefer to see animals and agriculture than concrete, cars and pollution."

"I am blessed, aesthetically by the green fields that surround Lincoln."

"... the aromas of the crops."; "Hearing cows and sheep at nighttime."

**Awareness and education (37, 21%)**
Being exposed and connected to farming practices (24, 13%)
Knowing the origin of food and its producing process (17, 10%)
Farming landscapes are something to be expected (4, 2%)
The opportunities to learn from growers (3, 2%)

"I have often missed working on farms as my job is in an office in the city and the farms of my youth are no longer in family hands. I wish my children have the same privilege I did growing up, to see and experience the freedom and hardships of people on the land, their courage and character."

"You know where your food comes from."

"I like being surrounded by rural people who are hands on and are genuine. They're great to learn from when growing my own garden."

**Being supportive (28, 16%)**
Support local growers and the community (20, 11%)
Positive image of farmers (10, 6%)

"I prefer to buy local where possible to support rural industries and townships."

"Farmers are salt of the earth folk. Hard working, love their land, their animals and lifestyles. They are generally positive people. Will do anything for someone who needs help etc. of cause I'm generalising, but it's how I feel."

**Associative benefits (23, 13%)**
Biophilic benefits provided by farm animals (13, 7%)
The opportunities to visit and get involved (6, 3%)
Self-sufficiency (2, 1%)

"It is so natural and great to see animals grazing in the fields and look happy..."

"I have kids that I would love to keep attached to farming... etc. I was lucky to grow up on a farm but they are not. Local areas could be a place my kids could be part of."

"Producing our own food is important to our family as we have interests in this area and value self-sufficiency and being connected with our environment."

**Socio-economic benefits (22, 12%)**
Sense of community (12, 7%)
Rural-urban connection (5, 3%)
Feed the population (2, 1%)
Employment opportunities (2, 1%)
Important for economy (2, 1%)

"Sense of community (positive neighbourhood relations)."

"Give opportunity for close rural/urban connection, a community that supports each other with fresh meat and produce."

"Concerned about loss of productive land to development. We need to be able to feed our own people and export excess for NZ economy."

"The encouragement of local food producing and employment."; "Helps local economy."

**Good land use (15, 8%)**

"I like seeing land being used productively, particularly if established trees and hedges are left in place. Psychologically more pleasing than sprawling housing areas."

**Environmental benefits (14, 8%)**
Environmental and ecological benefits (5, 3%)
Fresh air (5, 3%)
Less transport (5, 3%)

"I like the environmental benefits of supporting local food production."

"More open air around a farming area and being able to breath the fresher air."

"I like to see the food being grown and consumed locally as it means it's fresh and very little cost in dollars and to the environment, getting it to consumers."

**Personal connection to farming practices (11, 6%)**

"Most of people my age grew up not far from farms before urban areas started to spread out. I have no problem with farmers making a living in close proximity to where I live."

**Responsible farming practices (6, 3%)**

"I think farmer get a bad wrap from townies. Most/all farmers I know are good at what they do and do it as efficiently as possible, with a great deal of respect to the land and environment."

**Figure 8.** Respondents' stated reasons for their positive perceptions. Representative examples of relevant comments are presented alongside each group of reasons. All percentages shown above are relative to the 178 respondents who indicated their attitudes toward food-producing landscapes. Some respondents stated more than one reason. Note: themes commented by less than 1% of respondents were omitted from this figure.

To further examine the relationship between the predictor variables and residents' perception towards food-producing landscapes, a range of other statistical tests were conducted. A Spearman's rank correlation test was conducted to determine if there were any correlation relationships between the ordinal predictor variables (i.e., age, zone, number of years lived in this property, house age, land size, and personal connection to the farmers) and the outcome variable (i.e., residents' perception towards food-producing landscapes). As shown in Table 7, a two-tailed test of significance indicated that there were no significant correlations between the outcome variable and any of the predictor variables ($0.236 < p < 0.787$), suggesting that the age of the residents, how far away they live from food-producing landscapes, how long they have lived in their property, the age of their house, the size of their land, and their personal connection to local food producers have no significant impact on their attitudes toward food-producing landscapes close to their homes.

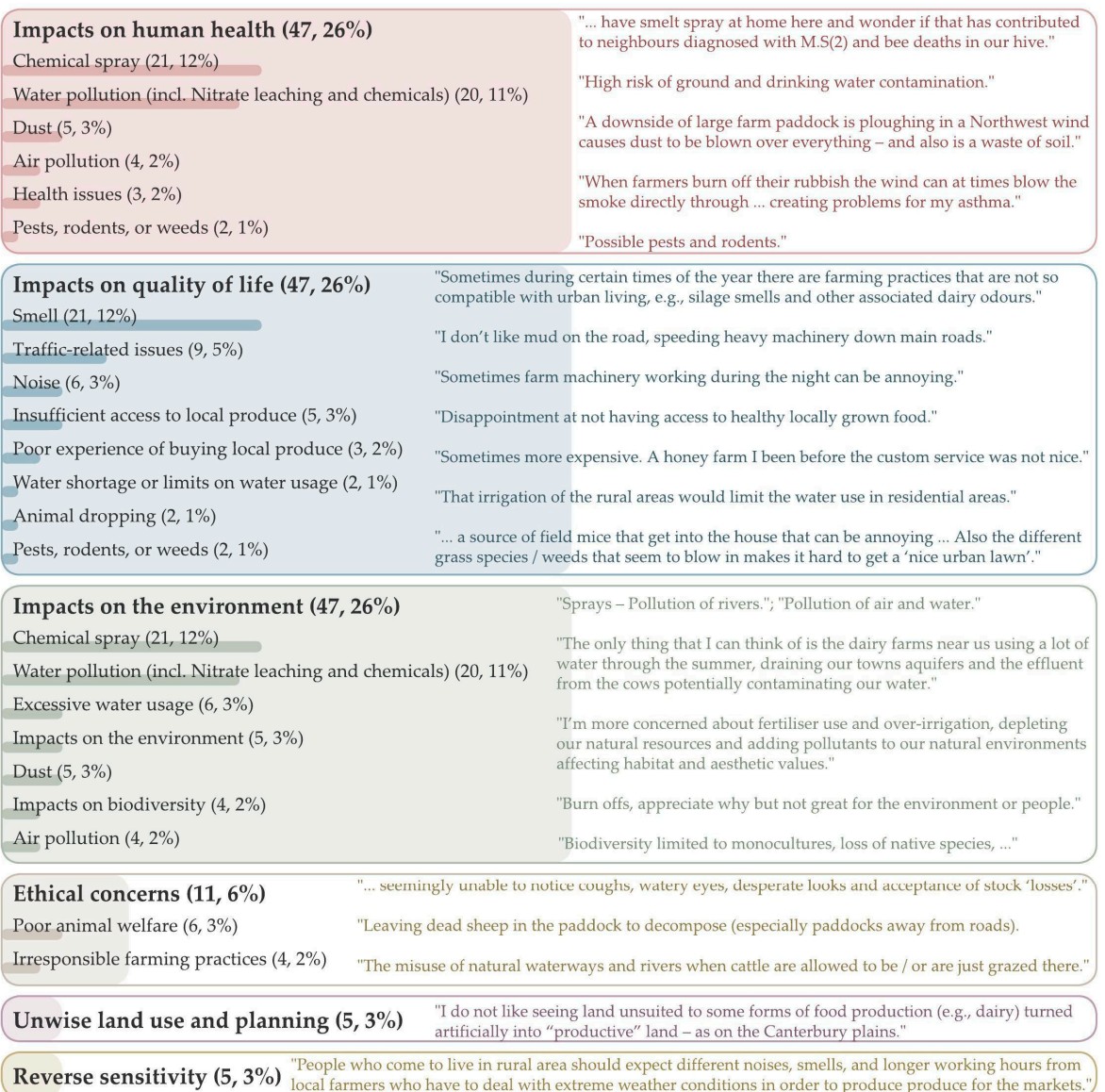

**Figure 9.** Respondents' stated reasons for their negative perceptions. Representative examples of comments are presented alongside each group of reasons. All percentages shown above are relative to the 178 respondents who indicated their attitudes toward food-producing landscapes. Some respondents stated more than one reason. Note: themes commented by less than 1% of respondents were omitted from this figure.

Mann–Whitney U tests were performed to evaluate whether residents' attitudes differed by the binary variables (i.e., respondents' gender, their property ownership status, farming background, adjacency to commercial farms, whether they produced food at home, and whether they knew any channels of purchasing local produce). The results of these statistical tests indicated that there were no significant inter-group differences for these six independent variables ($0.459 < p < 0.987$), suggesting that the tested factors make no significant statistical difference in their attitudes towards food-producing landscapes.

**Table 7.** Results for Spearman's rank correlation for the ordinal predictor variables and the outcome variable (residents' perception towards food-producing landscapes).

| Perception towards Food-Producing Landscapes | Age | Zone | Number of Years Lived in this Property | House Age | Land Size | Personal Connection to the Farmers |
|---|---|---|---|---|---|---|
| Correlation Coefficient | −0.04 | 0.02 | −0.09 | −0.08 | 0.05 | 0.05 |
| Sig. (2-tailed) | 0.573 | 0.787 | 0.236 | 0.288 | 0.555 | 0.479 |
| N | 176 | 178 | 175 | 176 | 171 | 174 |

Kruskal–Wallis tests were conducted to examine the residents' attitude differences among different groups of variables (i.e., household types, towns, and land titles). The tests for household types and land titles, which were corrected for tied ranks, were not significant (household types $\chi^2(4, N = 178) = 2.710$, $p = 0.607$; land titles $\chi^2(2, N = 174) = 2.173$, $p = 0.337$). These results suggest that the residents who live in different types of households (e.g., live alone, live with their partner, with their family or with flatmates) and different types of properties (residential sections, lifestyle blocks (see note 2) or farms) have no significant statistical differences in their attitudes toward food-producing landscapes close to their homes. However, the test for the town variable, which was adjusted for tied ranks, showed there was a significant difference in the residents' attitude between the different towns ($\chi^2(2, N = 178) = 7.354$, $p = 0.025$), with a mean attitude score rank of 82.33 for Darfield, 83.00 for Rolleston and 101.75 for Lincoln (as shown in Table 8).

**Table 8.** Descriptive statistics for the residents' attitude score per town group.

| Group | N | Mean | SD | Mean Rank |
|---|---|---|---|---|
| Darfield | 36 | 4.39 | 0.766 | 82.33 |
| Lincoln | 63 | 4.68 | 0.591 | 101.75 |
| Rolleston | 79 | 4.42 | 0.709 | 83.00 |

Post hoc comparisons were conducted to evaluate pairwise differences among the three town groups by using Mann–Whitney Tests, controlling for Type I error across tests by using the Bonferroni approach. As shown in Table 9, the difference between Darfield and Lincoln was statistically significant ($U(N_{Darfield} = 36, N_{Lincoln} = 63) = 889$, $z = −2.14$, $p = 0.032$). Similarly, the difference between Lincoln and Rolleston was also statistically significant ($U(N_{Lincoln} = 63, N_{Lincoln} = 79) = 1967$, $z = −2.51$, $p = 0.012$). No significant statistical difference was evident between Darfield and Rolleston ($U(N_{Darfield} = 36, N_{Rolleston} = 79) = 1409$, $z = −0.09$, $p = 0.928$). These comparison results suggest that Lincoln residents felt significantly more positive than both Rolleston and Darfield residents. This initial finding will serve as the focus for follow-up research to explore the reasons for this.

**Table 9.** Results of post hoc comparisons using Mann–Whitney Tests, with a Bonferroni adjustment.

| Sample 1-Sampe 2 | Mann-Whitney U | Z | Asymp. Sig. (2-Tailed) |
|---|---|---|---|
| Darfield-Lincoln | 889 | −2.14 | 0.032 |
| Darfield-Rolleston | 1409 | −0.091 | 0.928 |
| Lincoln-Rolleston | 1967 | −2.51 | 0.012 |

In summary, the results of the Spearman's rank correlation test, Mann–Whitney U tests, and Kruskal–Wallis tests suggested that among all the collected predictor variables, the only factor that had an impact on the residents' overall perception towards food-producing landscapes was the township in which they live. These results were comparable to those of the ordinal logistic regression models, suggesting the results are robust.

### 3.5.2. Perception towards Specific Types of Production Practices

Table 10 outlines residents' perceptions towards specific types of production practices within their local landscape. For all production types, the number of residents who felt positive was much greater than the number who felt negative. This corresponded with the pattern of overall perception illustrated in Table 6. The survey participants were given an 'N.A.' option, which allowed them to indicate that they were not aware of any farms nearby producing such food.

**Table 10.** Residents' perception towards specific types of production practices survey data.

| Production Practice | EP | MP | N | MN | EN | N.A. * | N |
|---|---|---|---|---|---|---|---|
| Dairy | 29 | 59 | 22 | 17 | 11 | 40 | 178 |
| Beef | 44 | 52 | 20 | 8 | 2 | 52 | 178 |
| Lamb | 56 | 55 | 14 | 5 | 2 | 46 | 178 |
| Crops for livestock consumption | 57 | 50 | 20 | 2 | 2 | 45 | 176 |
| Market garden products | 81 | 39 | 14 | 2 | 1 | 39 | 176 |
| Orchard fruits (excl. berries) | 58 | 28 | 11 | 1 | 1 | 75 | 174 |
| Field Crops for human consumption | 69 | 34 | 9 | 1 | 0 | 62 | 175 |
| Berries | 69 | 33 | 12 | 0 | 0 | 59 | 173 |
| Honey | 72 | 27 | 8 | 1 | 0 | 65 | 173 |

\* EP = Extremely positive; MP = Mostly positive; N = Neutral; MN = Mostly negative; N.A. = No known farms.

Figure 10 illustrates each perceptual score's percentage distribution within the population that rated each type of production practice, as well as the proportion of the group who rated that production type, and how many that did not. More than three quarters of the surveyed respondents knew a least one local market garden. The proportions for dairy farms, beef farms, livestock cropping and lamb farms were similar, at around three quarters. Local farms producing berries, honey and field crops for human consumption were slightly less recognisable, known by about two thirds of residents. Fruit orchards were the least recognisable production type, with a proportion slightly over half.

For respondents who rated the production types, dairy production received the most negative feedback with 17 (12%) responses rating their perception of local dairy production as 'mostly negative', and 11 (8%) rating it as 'extremely negative' (as shown in Figure 10). However, being the most negatively rated production type, dairy farming still received more positive feedback than negative, with 59 (43%) respondents rating it 'mostly positive' and 29 (21%) rating it 'extremely positive'. Following dairy, the other two types of livestock-based production systems, beef and lamb received the second and third most negative ratings, respectively. In comparison to the percentage of positive ratings they received (76% for beef and 84% for lamb), the percentage of their negative ratings (8% and 6%, respectively) was minor. Similarly, the perception towards livestock crops was predominantly positive, with only 2 (2%) respondents rating it 'mostly negative' and another 2 (2%) rating it 'extremely negative'. Apart from these four types of animal-related production types, the other five types were considered as positive landscape systems by most respondents (as shown in Figure 10), with no or very small percentages (less than 2%) of respondents indicating negative perceptions toward them.

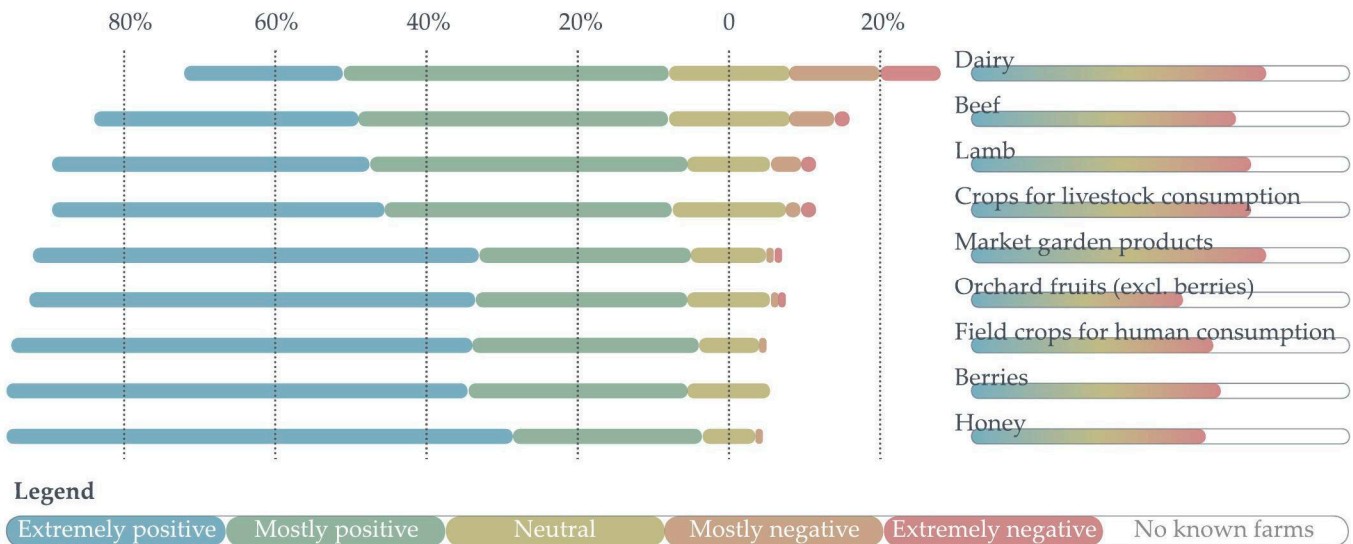

**Figure 10.** The bar charts on the left illustrate the percentage distribution of the respondents' perceptual scores towards a specific type of production practice (all proportions are relative to the number of respondents who indicated they were aware of at least one local farm producing that specific type of produce and thereby were able to indicate their perceptions toward that type of production practice). The bar chart on the right illustrates the percentage of respondents who were aware of at least one local farm of that specific type.

To develop a thorough comprehension of the respondents' sentiments toward each of these product types, participants were asked to provide a brief explanation of the reasons why they liked or did not like each of the food-production systems they rated. Their explanations were coded as per the method explained in Section 2.1.4. The reasons for feeling positive and negative that were extracted from the explanations were then cross tabulated with the nine production types.

Figure 11 presents the residents' stated reasons for feeling positive towards each of the nine types of food production. Overall, there were 95 respondents who specified their reasons for positive perception. Among the 95 responses, berries were the production type that was mentioned most, by 23 respondents. One of the key contributors to their positive perceptions towards this production type were the produce-related benefits, with many respondents stating that berry products can be easily accessed locally by 'Pick-Your-Own' (PYO). This was also echoed by several respondents who stated other associative benefits of berry production, such as the opportunity to visit berry farms (which were often stated as a 'fun' experience). Berry farms were also considered a good way of using the land by six respondents. Figure 11 illustrates similar reasons for respondents feeling positive toward market garden production, except for a lower number of associative benefits identified, indicating that market garden production is less participatory than berry production. Apart from berries and market garden produce, honey was also mentioned by six respondents for its produce-related benefits, with most comments relating to the benefits of local accessibility.

It is noteworthy that in many cases respondents provided their reasons without specifically referring to a particular production type. Instead, explanations were kept at a 'general' level, commenting on food-producing landscapes 'as a whole'. As shown in Figure 11, among the 95 respondents, 63 provided comments that were not specifically related to any individual product type. The most mentioned reasons for feeling positive toward all product types 'as a whole' were the socio-economic benefits, which were expressed by ideas such as 'we have to use these productive lands to feed the population', 'food production is important for the economy' and 'local food production prevents reliance on imported food'. In addition, as mentioned by 11 respondents, awareness and education

were also contributors to the generally positive perceptions of residents. They reasoned that having farming landscapes visible provides them and others with the opportunity to see and connect with farming practices. It was also stated that allowing farming to be visible helps the public to know where their food comes from. Similarly, the perceptual benefits (including rural outlook and lifestyle), responsible farming practices, being supportive and personal connections to farming were also key factors stated that contributed positively to residents' perception toward food-producing landscapes.

| Reasons for feeling positive | Not specified | Dairy | Beef | Sheep or lamb | Market garden products | Orchard fruits | Berries | Honey | Field crops for human | Crops for livestock | Total |
|---|---|---|---|---|---|---|---|---|---|---|---|
| Produce-related benefits | 11 | 2 | 1 | 0 | 10 | 3 | 11 | 6 | 1 | 2 | 29 |
| Environmental benefits | 2 | 0 | 0 | 0 | 0 | 1 | 1 | 1 | 0 | 0 | 4 |
| Associative benefits | 4 | 2 | 1 | 2 | 1 | 2 | 7 | 1 | 3 | 3 | 11 |
| Perceptual benefits | 8 | 0 | 0 | 0 | 0 | 0 | 0 | 0 | 0 | 0 | 8 |
| Socio-economic benefits | 18 | 0 | 0 | 1 | 1 | 0 | 1 | 0 | 0 | 0 | 21 |
| Awareness and education | 11 | 1 | 1 | 0 | 1 | 1 | 1 | 0 | 1 | 1 | 12 |
| Good land use | 5 | 3 | 3 | 3 | 7 | 5 | 6 | 2 | 6 | 5 | 12 |
| Good farming practice | 8 | 3 | 1 | 2 | 1 | 1 | 1 | 1 | 1 | 1 | 12 |
| Being supportive | 7 | 1 | 1 | 0 | 1 | 1 | 1 | 2 | 0 | 2 | 9 |
| Personal connections to farming practices | 6 | 2 | 1 | 0 | 1 | 0 | 0 | 0 | 0 | 1 | 8 |
| Others | 9 | 1 | 1 | 1 | 2 | 2 | 0 | 3 | 0 | 2 | 13 |
| **Total** | 63 | 11 | 8 | 9 | 21 | 11 | 23 | 14 | 10 | 13 | **95** |

**Figure 11.** Residents' reasons for feeling positive towards different production types.

Figure 12 presents residents' stated reasons for feeling negatively towards each of the nine production types. These perceptions included impacts on human health and quality of life, impacts on the environment, inappropriate land use and ethical concerns (particularly regarding the perceived negative impacts of dairy farming on the environment). Unlike the pattern for positive perception, there were few instances where respondents explained their reasons for feeling negative without specifying production types, as shown in Figure 12. This clear contrast indicates that most surveyed residents felt positively toward food-producing landscapes 'as a whole' and believe that those landscapes offer various benefits to residents regardless of the type of production. The survey elicited, however, a small number of residents who had specific negative perceptions towards particular production types. This indicates that it may be possible to manage the perceived negative impacts that food-producing landscapes have on resident properties by directing the type of production occurring within proximity to housing.

Figure 12 also reveals a strong pattern where almost all negative perceptions of food-producing landscapes are associated with the three types of livestock-based production types. Among the 35 respondents who stated reasons for their negative perceptions, 31 specifically referred to dairy production, 14 referred to beef, and 9 referred to sheep or lamb. The impacts that these production types have on the environment were the most stated reason for the negative perception, with specific issues including water pollution, excessive water usage and greenhouse gas emissions. Following that, impacts on human health and quality of life were also key concerns identified for livestock-based production types. These impacts were largely related to water pollution caused by nitrogen leaching, with a smaller number of respondents commenting on chemical spray drift and smells.

Ethical concerns related to animal welfare were also identified as a key contributor to the negative perceptions toward livestock-based production types.

Additionally, there was a small number of comments relating to plant-based production systems and their associated perceived negative impacts on the environment and human health. Most of these comments referred specifically to the use of chemicals and the impact of spray drift.

| Reasons for feeling negative | Production types | Not specified | Dairy | Beef | Sheep or lamb | Market garden products | Orchard fruits | Berries | Honey | Field crops for human | Crops for livestock | Total |
|---|---|---|---|---|---|---|---|---|---|---|---|---|
| Impacts on human health and quality of life | | 1 | 22 | 11 | 6 | 2 | 1 | 0 | 0 | 1 | 2 | 26 |
| Impacts on the environment | | 2 | 27 | 12 | 6 | 1 | 1 | 0 | 0 | 1 | 4 | 32 |
| Unwise land use | | 0 | 6 | 4 | 1 | 0 | 0 | 0 | 0 | 0 | 0 | 6 |
| Ethical concerns | | 0 | 7 | 3 | 4 | 0 | 0 | 0 | 1 | 0 | 0 | 7 |
| **Total** | | 2 | 31 | 14 | 9 | 2 | 1 | 0 | 1 | 1 | 4 | 35 |

**Figure 12.** Residents' reasons for feeling negative towards different production types.

3.5.3. Perception towards Specific Production Approaches

Regarding the production approach, the number of residents who felt positive was again much more significant than the number who felt negative, as illustrated in Table 11. Similarly, the participants were given an 'N.A.' option which allowed them to indicate that they did not know this farming approach and therefore did not rate it.

**Table 11.** Residents' feelings towards different growing/farming approaches within the peri-urban zone.

| Approach | EP | MP | N | MN | EN | N.A. | N |
|---|---|---|---|---|---|---|---|
| Conventional farming | 61 | 75 | 27 | 8 | 2 | 5 | 178 |
| Organic farming | 87 | 62 | 22 | 3 | 0 | 3 | 177 |
| Regenerative farming | 75 | 44 | 19 | 2 | 0 | 33 | 173 |
| Community garden | 98 | 53 | 22 | 1 | 0 | 4 | 178 |
| Mahika kai/Mahinga kai and Māri kai (see note 3) | 49 | 18 | 15 | 0 | 1 | 85 | 168 |

Note: EP = Extremely positive; MP = Mostly positive; N = Neutral; MN = Mostly negative; N.A. = do not know this farming approach.

Figure 13 shows each perceptual score's percentage distribution within the population that rated the farming approach, as well as the proportion of the group who rated it and who did not. Almost all respondents had knowledge of conventional farming approaches, organic farming and community gardens. Regenerative farming was known by slightly fewer residents. Mahika kai/Mahinga kai and Māra kai were the least well-known farming approaches, with more than half of surveyed residents not familiar with these traditional Māori concepts/approaches.

Within the groups of respondents who rated the farming approaches, conventional farming received the most negative feedback (as shown in Figure 13), with 8 (5%) responses rating them 'mostly negative' and 2 (1%) 'extremely negative'. However, the conventional

farming approach still received many more positive ratings than negative, with 75 (43%) respondents rating it 'mostly positive' and 61 (35%) rating it 'extremely positive'. Apart from conventional farming, other approaches were all considered as positive landscape systems by most respondents, with only very small percentages (less than 2%) of respondents indicating negative perceptions toward them.

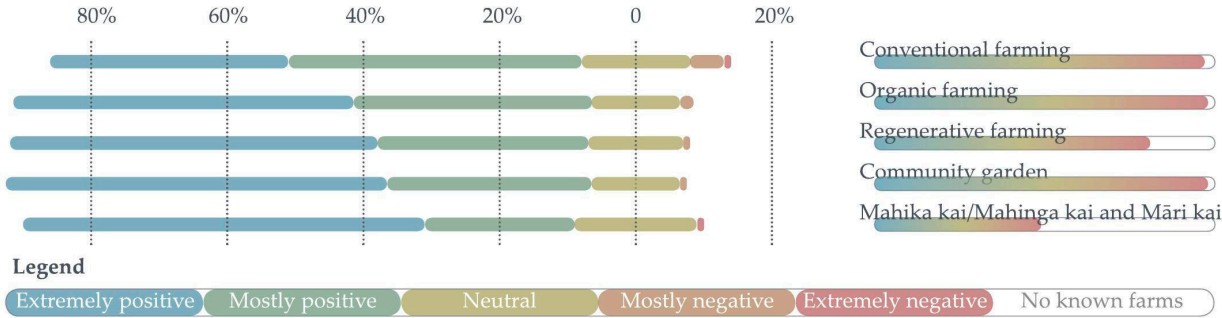

**Figure 13.** The bar charts on the left illustrate the percentage distribution of the respondents' perceptual scores towards a specific type of farming approach (all proportions are relative to the number of respondents who indicated they had knowledge of that specific type of farming approach and thereby be able to indicate their perceptions toward it). The bar chart on the right illustrates the percentage of respondents who had knowledge of that specific farming approach.

*3.6. Food Producers' Survey*

Overall, the food producer survey yielded 11 responses in total, of which 10 were accepted and one was rejected due to an incomplete consent form. The overall response rate was 20%, as 50 surveys were distributed.

Table 12 outlines the demographic profile of the respondents and their property characteristics. Overall, the respondent population for this survey was relatively gender-balanced and evenly distributed across age groups. Sixty per cent of respondents own the farms they produce food on. All surveyed producers had farmed on their existing land for more than 4 years, and 40% had farmed on that land for 19 years or more. Most farms were sized between 5 and 50 hectares, with two farms sized between 1 and 5 hectares, and one smaller than 1 hectare. Sixty per cent of respondents live on the farm they produce food on. Regarding the land uses bordering the respondents' farms, 33% of farms directly border residential land. All surveyed food producers operate within 10 km of residentially zoned land. Figure 14 depicts a typical greenfield residential subdivision, being developed within peri-urban farmland within the Waikirikiri Selwyn District.

**Table 12.** Food producers' demographic and property characteristics survey data. All percentages shown below are relative to the number of respondents who provided a valid answer to that question.

| Attribute | Description | Count | N | Percentage |
|---|---|---|---|---|
| Age bracket | 18–29 years old | 0 | | 0.00% |
| | 30–39 years old | 2 | | 20.00% |
| | 40–49 years old | 1 | 10 | 10.00% |
| | 50–64 years old | 3 | | 30.00% |
| | 65+ years old | 4 | | 40.00% |
| Gender | Female | 4 | | 40.00% |
| | Male | 5 | 10 | 50.00% |
| | Non-binary | 1 | | 10.00% |
| Property ownership | Own | 6 | | 60.00% |
| | Lease | 3 | 10 | 30.00% |
| | Mix | 1 | | 10.00% |

| Attribute | Description | Count | N | Percentage |
|---|---|---|---|---|
| Number of years farming on their current land | Less than one year | 0 | | 0.00% |
| | 1–4 year | 0 | | 0.00% |
| | 4–7 year | 2 | | 20.00% |
| | 7–10 year | 2 | 10 | 20.00% |
| | 10–13 year | 1 | | 10.00% |
| | 13–16 year | 1 | | 10.00% |
| | 16–19 year | 0 | | 0.00% |
| | 19 years or more | 4 | | 40.00% |
| Size of the farm | Less than 1 ha | 1 | | 10.00% |
| | 1–5 ha | 2 | | 20.00% |
| | 5–10 ha | 2 | 10 | 20.00% |
| | 10–20 ha | 3 | | 30.00% |
| | 20–50 ha | 2 | | 20.00% |
| | More than 50 ha | 0 | | 0.00% |
| Do you live on the land that you produce food on? | Yes | 6 | 10 | 60.00% |
| | No | 4 | | 40.00% |
| Distance to the closest residential property | Residential property borders my farm | 3 | | 33.33% |
| | Less than 1 km | 1 | | 11.11% |
| | Less than 5 km | 3 | 9 | 33.33% |
| | Less than 10 km | 2 | | 22.22% |
| | No residential property within 10 km | 0 | | 0.00% |
| Current land use of the bordering landscapes (some farms border more than one type of land use) | Residential/subdivision lots | 3 | | 33.33% |
| | Lifestyle blocks (see note 2) | 7 | 9 | 77.78% |
| | Farms | 6 | | 66.67% |
| | Public reserves or other natural landscapes | 1 | | 11.11% |

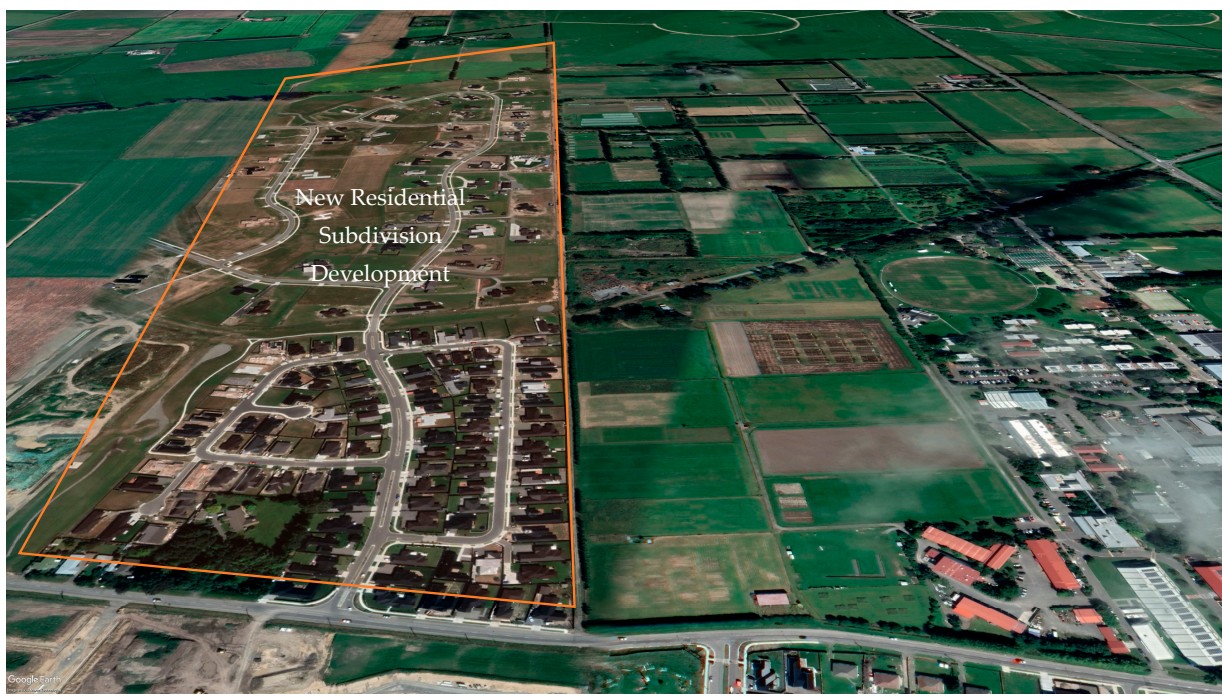

**Figure 14.** Typical development of a greenfield residential subdivision on peri-urban farmland, on the outskirts of Lincoln (Google Earth Pro V 7.3.6.9345. (26 August 2018). Lincoln, New Zealand. 43°38′58.92″ S, 172°27′54.37″ E, Eye alt 427 m. Image Landsat/Copernicus, Data SIO, NOAA, U.S. Navy, NGA, GEBCO. Airbus 2023, Maxar Technologies 2023. http://www/earth/google.com (accessed on 19 August 2023).

Table 13 illustrates the behavioural characteristics of the surveyed food producers, with half of the respondents producing market garden products (vegetables), 40% producing non-berry fruits, 30% producing honey and a slightly smaller proportion of respondents producing beef, lamb, berries, livestock crops or field crops for human consumption.

**Table 13.** Food producers' behavioural characteristics and awareness survey data. All percentages shown below are relative to the number of respondents who provided a valid answer to that question.

| Attribute | Description | Count | N | Percentage |
|---|---|---|---|---|
| Type of food produced | Dairy | 0 | | 0.00% |
| | Beef | 2 | | 20.00% |
| | Lamb | 1 | | 10.00% |
| | Crops for livestock consumption | 1 | | 10.00% |
| | Market garden products | 5 | 10 | 50.00% |
| | Orchard fruits (excl. berries) | 4 | | 40.00% |
| | Field crops for human consumption | 1 | | 10.00% |
| | Berries | 1 | | 10.00% |
| | Honey | 3 | | 30.00% |
| | Other | 4 | | 40.00% |
| What market do you produce food for? | International markets | 2 | | 22.22% |
| | Domestic supermarket | 1 | | 11.11% |
| | Local supermarket | 2 | | 22.22% |
| | Local independent stores | 3 | 9 | 33.33% |
| | Farm gate | 2 | | 22.22% |
| | Farmers' market | 7 | | 77.78% |
| | Other | 5 | | 55.56% |
| Farming approach | Conventional | 4 | | 44.44% |
| | Organic | 2 | 9 | 22.22% |
| | Other | 3 | | 33.33% |
| Do you want to stay farming in your current location? | Yes | 7 | | 87.50% |
| | Would like to continue somewhere else | 0 | 8 | 0.00% |
| | No, would like to stop farming | 1 | | 12.50% |
| Green waste treatment | Green bin | 0 | | 0.00% |
| | Composting system | 7 | 9 | 77.78% |
| | Animal feed | 4 | | 44.44% |
| | Other | 1 | | 11.11% |
| Approximate percentage of income generated from farming | 10% | 2 | | 22.22% |
| | 25% | 3 | 9 | 33.33% |
| | 50% | 3 | | 33.33% |
| | 100% | 1 | | 11.11% |

Many of the surveyed farms had a diverse range of products, with over 50% producing more than one type of produce. The surveyed participants also produced eggs, olive-based products, hazelnuts and pork. Apart from selling their produce in Farmers' Markets, a smaller proportion of respondents also reported selling their products through international markets, domestic and local supermarkets, local stores, farm-gate sales, sales direct to local restaurants, cafes and other hospitality outlets, the internet, home delivery and vegetable box schemes. Around half the respondents applied a 'conventional farming' approach, while the other half operated organically, spray-free and/or biologically. All surveyed food producers generated some income from the products they grew. One participant generated their whole income through produce sales, while the remaining participants generated 50% or less.

To understand the growers' and farmers' attitudes toward operating their food-producing enterprises within the peri-urban zone, the survey asked participants to answer how positively or negatively they felt about farming near residential neighbours. Different from the residents' predominantly positive perceptions towards farming landscapes, more

than half of the food producers felt neutral (55.6%) towards having residential neighbours around their farms (as shown in Table 14). While one third of the surveyed producers felt 'mostly positive', none felt 'extremely positive'. On the contrary, one respondent felt extremely negative towards farming near residential areas.

**Table 14.** Food producers' perception towards operating their food-producing enterprises within the peri-urban zone. All percentages shown below are relative to the nine respondents who indicated their attitudes.

| Attribute | Description | Count | N | Percentage |
|---|---|---|---|---|
| Overall perception towards producing food in the peri-urban zone | Extremely positive | 0 | | 0.00% |
| | Mostly positive | 3 | | 33.33% |
| | Neutral | 5 | 9 | 55.56% |
| | Mostly negative | 0 | | 0.00% |
| | Extremely negative | 1 | | 11.11% |

Participant food producers were asked to provide a brief explanation of anything they liked or disliked about farming near residential neighbours. Their comments were coded as per the method explained in Section 2.1.4. Figure 15 presents the respondents' stated reasons for their positive perceptions, along with some representative comments. The most frequently cited reason for positive perceptions was the proximity to customers. Respondents felt that this made their sales easier and reduced the transport cost both financially and environmentally. Two respondents explained that farming in the peri-urban zone provided the opportunity to get the community involved and to improve their interaction with the public. One respondent acknowledged that farming in the peri-urban zone means that they are close to shops and services, and at the same time they also benefit from adjacent residential infrastructure improvements such as the fibre network.

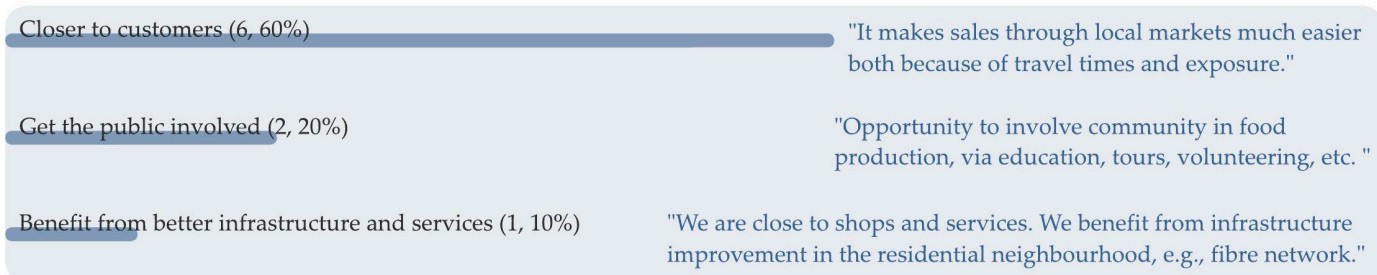

**Figure 15.** Food producers' stated reasons for their positive perceptions towards having residential areas near their farms. Representative examples of comments are presented alongside each reason. All percentages shown above are relative to the 10 valid responses. Some respondents stated more than one reason, and some stated none.

Figure 16 provides an overview of the respondents' reported reasons for their negative perceptions, accompanied by representative comments. The biggest concern food producers had with farming in the peri-urban zone was the number of complaints received from their residential neighbours. Just as some residents had concerns about the use of chemicals, some organic or spray-free producers were also concerned about the spray drift that comes from their residential neighbours. There were also two respondents concerned with security. There were also less-cited concerns mentioned, including noise from residential neighbours, excessive water usage by lifestyle blocks (see note 2) and neighbourhood cats.

In summary, different from the residents' predominantly positive perceptions towards food-producing landscapes, food producers presented a response that portrayed a more neutral perception towards farming near their residential neighbours.

Complaints and judgements from neighbours (5, 50%)

"Complaints from neighbours re: Aesthetics (our property not perfectly manicured); Tree height of existing trees; Stock noises and smells; Machine noise (we are currently mitigating this by operating machinery within certain hours).."

Spray drift (2, 20%)

"We had an incident in the past 6 wks where one of our new neighbours (from the recent subdivision on our southern boundary) sprayed Roundup on the LU side of their boundary fence, next to the BHU organic farm. "

Break-ins (2, 20%)

"Concerns about fence jumping + entering private property without consent, which could lead to injury, theft, damage, whether malicious intent or not. "

Noise (1, 10%)

"Noise: Otherwise, peaceful days disrupted by neighbours outdoor music systems blasting music. "

Cats (1, 10%)

"Cats!! So many cats! Attacking native bird species. Scaring/attacking chickens. Leave 'manure'. "

Excessive water usage of lifestyle blocks (1, 10%)

"I do not like 'lifestyle' blocks being used by large farmers as an excuse for excessive water usage with no return, according to them. "

Have no negative experiences (1, 10%)

"We haven't had any negative experiences. "

**Figure 16.** Food producers' stated reasons for their negative perceptions towards having residential areas near their farms. Representative examples of relevant comments are presented below each reason. All percentages shown above are relative to the 10 valid responses. Some respondents stated more than one reason, and some stated none.

## 4. Discussion

This case study has illustrated community perspectives of peri-urban food production. It included perspectives from both residents living within the peri-urban zone of Ōtautahi Christchurch and those of food producers operating within it. The research aimed to explore perceptions of peri-urban food production in the face of unprecedented pressure for land-use change from urban expansion and increasing levels of 'reverse sensitivity'. Through surveying the peri-urban community this research explored the positive and negative perceptions of peri-urban food production to further understand how future planning could address the competing needs of these two land uses, and enable the continuation of food production alongside housing.

The findings revealed that, overall, residents highly value food-production landscapes, with 92.13% (164) of respondents feeling either 'extremely positive' or 'mostly positive' towards food production occurring within their local vicinity. Just 1.69% (3) of respondents felt 'negatively', with the remaining 6.18% (11) feeling neutral. The overwhelmingly positive perception was not correlated with demographic or property attributes, with residents feeling positive toward food-producing landscapes regardless of their individual circumstances. Contrastingly, food producers did not feel as positively as residents, with 66.66% (6) of food producer respondents feeling either 'neutral' (55.56%, 5) or 'negative' (1, 11.11%).

The surveyed participants identified multiple benefits and hindrances related to having food produced close to where they live. The largest perceived benefit of peri-urban food production was access: *"Lots of fresh locally grown food and veges are easily accessible, at both markets and gate sales. Often these are cheaper than [the] supermarket",* and *"It is amazing to have access to such a wide range of produce".* However, the issue of the difficulty of accessing local produce was also highlighted as an issue for some participants, illustrated

by comments such as *"Disappointment at not having access to healthy locally grown food"*. For future peri-urban planning consideration, the issue of inconsistent accessibility across the peri-urban zone might be addressed through policy and planning that permit more, or more evenly distributed local sales outlets (e.g., farm-gate sales, 'pick your own' and local markets), allowing more residents easy access to the produce being grown within their local landscape.

The category of 'rural outlook' to farming landscapes and the associated lifestyle benefits of this was also a strong narrative within the survey responses. Comments such as *"I like the openness of the rural landscape, the feeling of being part of 'traditional' New Zealand country life"*, and *"We came to Lincoln to be rural, have the wide open spaces and fresh air. We much prefer to see animals and agriculture than concrete, cars and pollution"* typified many narratives. The survey also elicited many 'associative' benefits of living near a food-production landscape with one participant stating *"I like being surrounded by rural people who are hands on and are genuine. They're great to learn from when growing my own garden"*. Similarly, common among producers' perspectives was the acknowledgement of producing food close to customers. The opportunity for public involvement in farming and an appreciation of the associated benefits of closer proximity to good infrastructure services were noted, with one producer participant commenting *"It makes sales through local markets much easier both because of travel times and exposure"*. For future peri-urban planning, this research has highlighted that retaining the rural character and providing spaces for resident/producer engagement is important to building a strong peri-urban community. Both resident and producer participants identified benefits to living close to one another, with land-use planning providing a mechanism to support this going into the future, for example, through increased rural visibility, community gardens and farm open days.

Negative associations with food-production landscapes were also identified by both residents and producers. For resident participants, these negative perceptions generally fell into three main categories, including negative impacts on human health, negative impacts on 'quality of life' and negative impacts on the environment. An interesting finding of the survey was that respondents generally did not provide specific reasoning for their positive feelings towards peri-urban food production, as shown by such comments as *"[I] like having fresh food available to purchase keeping me healthy and strong while studying"*. In contrast, however, when asked to comment on negative aspects of living near food-producing landscapes, participants were more specific in their detail, for example, *"dairy farms near us [use] a lot of water through the summer, draining our towns aquifers and the effluent from the cows potentially contaminating our water"*. Attitudes relating to negative impacts on human health were represented by comments such as *"When farmers burn off their rubbish the wind can at times blow the smoke directly through our neighbourhood creating problems for my asthma"*, and *"a downside of large farm paddock is ploughing in a Northwest wind causing dust to be blown over everything—and also is a waste of soil from the paddock"*. With reference to negative impacts on the quality of life for residents, one participant stated that *"Sometimes during certain times of the year there are farming practices that are not compatible with urban living, e.g., silage smells and other associated dairy odours"*, and another stated that *"Sometimes farm machinery working during the night can be annoying especially if going all night"*. For perceived impacts on the environment, water pollution, sprays, fertiliser use and overuse of water were all mentioned. One resident commented, *"I'm more concerned about fertiliser use and over-irrigation depleting our natural resources and adding pollutants to our natural environments affecting habitat and aesthetic values."* Furthermore, ethical concerns relating to animal welfare were included by some. For example, *"[farmers are] ...seemingly unable to notice coughs, watery eyes, desperate looks and acceptance of stock 'losses'.* Although there were negative factors of living near food-producing landscapes mentioned throughout the residents' survey, overall, they were deemed acceptable to most participants. The negative impacts were largely considered as negligible or tolerable 'side-effects' of rural living. Most residents acknowledged the fact that farming came first, so living here was considered a decision made after weighing the potential impacts, ultimately perceiving that

the overall benefits that living near a food-producing landscape can provide are greater than the overall perceived negative impacts.

Overall, the food producers surveyed felt less positively towards having residential land adjacent to their farms. This leads to the question—how can spatial land-use planning support the negotiation and mediation of land-use values and function within the peri-urban zone, to support the needs of food producers as well as those of residents? When considering future peri-urban planning solutions to address the reported issues and imbalance of satisfaction levels between the two surveyed populations, several resolution strategies have been highlighted by this research. As touched upon above, raising the level of education and agricultural literacy within the resident population was one idea highlighted through the food producer survey, so that resident neighbors better understand farming cycles and practices. From a planning perspective, this could be addressed by raising the level of visibility of farming and providing visual and physical access to farms through the seasons, with residents learning the cycles and necessities of growing and raising food. From a resident perspective, 'buffering' farms from neighboring residential properties was also commented upon. The use of green belts was a suggested possibility to mediate some immediate perceived negative impacts of farming on residential neighbors. Separating production types deemed more negative from areas of housing could also support a reduction in reverse sensitivity. Specific production types, such as dairy farming, were perceived more negatively than others. Setting land-use policy to determine the spatial location of certain production types may help alleviate perceived negativities and support the continuation of those production types deemed to be more conducive to urban neighbours.

Negotiating land-use demands within the peri-urban zone, particularly between housing and food production, is a growing area of concern for Aotearoa New Zealand, as it is in many countries globally. In recent decades, the rural–urban dichotomy has been exacerbated by land-use zoning policies that have sought to separate land uses. The findings of this research, however, have shown the multiple identified positive aspects of having food production occurring close to residential communities. The exploration of specific land-use typologies and relationships to reduce the identified negative impacts will be the focus of subsequent work.

## 5. Conclusions

As war, pandemics and climate change bring renewed concerns over urban food security and resilience, cities globally are re-thinking their spatial relationship with food-producing landscapes. Re-prioritising food production over low-density residential sprawl as an important and necessary land-use function close to cities is essential to ensure holistic urban resilience in the future. Such a re-prioritisation of food production and access is then considered alongside adequate shelter (housing), clean water and fresh air.

Having been left off the urban planning agenda for decades, food production on the edges of cities in Aotearoa New Zealand, we contend, is an essential component of long-term urban resilience. Understanding the issues and attitudes of both residents and food producers in this zone will help ensure appropriate and effective land-use planning and design to mitigate potential negative effects, while enhancing the positive outcomes of peri-urban food production.

This research has been conducted as part of an ongoing study into peri-urban food production. This paper presents the issues identified by two key stakeholder groups, with future research building on this to explore potential land-use typologies, and 'test' alternative spatial planning scenarios.

**Author Contributions:** Conceptualization, S.D. and G.C.; methodology, S.D. and G.C.; software, G.C.; validation, G.C.; formal analysis, G.C.; data curation, G.C.; writing—original draft preparation, S.D., G.C. and N.D.; writing—review and editing, S.D.; project administration, S.D.; funding acquisition, S.D. All authors have read and agreed to the published version of the manuscript.

**Funding:** This research was funded by the Our Land and Water National Science Challenge, New Zealand (grant number 2022CF003).

**Data Availability Statement:** The data presented in this study are available on request from the corresponding author. The data are not publicly available due to privacy requirements of the participants.

**Acknowledgments:** Donald Royds from the Lincoln University School of Landscape Architecture generously undertook the drone photography shown in Figure 1 of this paper. The research presented in this paper is part of a larger study investigating peri-urban land use. Contributors to the wider study who have given their time for tasks related to the broader research include Professor Pablo Gregorini, Richard Morris and Marcus Robinson.

**Conflicts of Interest:** The authors declare no conflict of interest.

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
