# Peer review of "Housing and Food Production: Resident and Grower Perceptions of Peri-Urban Food-Production Landscapes"

_land, doi:10.3390/land12122091_

Round 1

Reviewer 1 Report (Previous Reviewer 2)

Comments and Suggestions for Authors I read the revised manuscript and I approve its acceptance. However, I found that the version I got is full of grammatical faults. And more important, that the paper is still read as a research report bringing too much information in a very un-appealing way. All the best,

Comments on the Quality of English Language I found that the version I got is full of grammatical faults.

Author Response

Thank you for your review - please see the attachment for our responses. 

Reviewer 2 Report (Previous Reviewer 3)

Comments and Suggestions for Authors

This paper is not ready yet for review. There are many sloppy expressions that must be improved. 

Abstract

line 23

The abstract needs to be shortened. It should flow well. English must be revised, and language must be smooth.

Very sloppy writing, for example, line 44 in the Introduction, lines 55-58 and lines 59-65.

Very difficult paragraph to follow (lines 66-74).

This paper is not ready for a review. For example, see line 150. Such mistakes are all over the paper.

Fig. 3. LUC Land use classification class should be the title and

1.      Versatile multiple land use

2.      ..

3.      ..

4.      ..

5.      ..

6.      ..

7.      ‘’

8.      Non-arable land

Run-on sentences and are observed from lines 696-707.

This paper has some good contents, but the write-up needs a through overhauling to make the flow smooth.

Comments on the Quality of English Language

Needs overhauling. 

Author Response

Thank you for your review - please see the attachment for our responses. 

Reviewer 3 Report (New Reviewer)

Comments and Suggestions for Authors

The manuscript is very good and well structured. However, it represents something which is already known in countries of Central and Eastern Europe. Namely, in all these ex-socialist countries the local government allowed using of municipally owned public land for growing the vegetables and other types of gardening until the public land will not be transmitted into the construction area.  Nevertheless, it might me interesting for the readers of West Europe, USA, and Canada, particularly considering the Sustainable Food Systems and Urban Agriculture. However, I kindly ask the authors to pay attention about writing the metric units. For instance,  on page 4/39 in line 147 you have 20km instead of 20 km and also on the same page in line 159, you have  22km instead of 22 km. Please use the spacebar check the whole text. Thank you.

Author Response

Thank you for your review - please see the attachment for our responses. 

Reviewer 4 Report (New Reviewer)

Comments and Suggestions for Authors

The authors designed the article to explores the perceptions and attitudes of both peri-urban residents and food producers, living and working within the peri-urban zone”. This survey sought to understand residents’ perceptions toward food-production landscapes close to their homes, as well as resident experiences with different production types and approaches but there are some discrepancies in the article which needs to be improved before final publication. The comments are added in the attached file.

Specific Comments

The topic is sound. There are many typographical and writing errors, I should say full of such errors. Please make correction throughout the article. However, the efforts have been made by the authors to make the findings relevant for an international audience. The manuscript holds potential to provide a good contribution to the Journal, with some corrections.

Comments on the Quality of English Language

The English needs to be rewrite as It has many mistakes, repeated words, poorly written sentences (also mentioned in the file). Some of the paragraphs should be written clearly.

Author Response

Thank you for your review - please see the attachment for our responses. 

Round 2

Reviewer 2 Report (Previous Reviewer 3)

Comments and Suggestions for Authors

The authors have addressed my concerns but the quality of responses could have been better. More specifically, the following issues are yet to be addressed including the edition of English to make the flow better. 

******************************************

Like in Tables 5 and 6, in pages 16 and 18 make clear how the percentages were calculated under different subheadings.

Figures 10 and 13 need to be customized.

 Page 26, line 671 percentage total does not add up to 100%

Figure 16 is very unclear. 

Comments on the Quality of English Language

English needs to be re-edited to make the flow better. 

Author Response

Point 1: In Tables 5 and 6, in pages 16 and 18 make clear how the percentages were calculated under different subheadings.

Response: Thanks for pointing out these possible confusing points. We’ve now added some notes to the captions of Tables 3-6 and Figures 8 and 9 to explain how the percentages were calculated.

Point 2: Figures 10 and 13 need to be customized.

Response: We’ve now rewritten the captions for Figures 10 and 13 to further clarify the ideas illustrated by those figures. We are not exactly sure what your recommendation is in relation to ‘customizing’, but hopefully our rewritten captions can address your concerns about these two diagrams.

Point 3: Page 26, line 671 percentage total does not add up to 100%.

Response: Thank you for bringing this to our attention. We’ve now added a note to explain that some farms border more than one type of land use, so that the percentage total does not add up to 100%.

Point 4: Figure 16 is very unclear.

Response: We’ve now added some further explanation to the captions of Figures 15 and 16 to explain the information contained within those figures, as well as how the percentages were calculated. However, it is worth noting that the total number we had in Table 14 is not the same as the total number in Figures 15 and 16. The reason for this difference is that one of the respondents didn’t answer the multi-choice question asking about their perception (presented in Table 14), but answered the short answer questions about what they like or dislike. So the total number in Table 14 is 9, but having 9 as the total number does not make sense for Figures 15 and 16, as the person that we excluded in Table 14 provided his/her answer for the questions of Figures 15 and 16.

Point 5: English needs to be re-edited to make the flow better.

Response:  We have had a senior Professor with a substantial international publication record (over 200 publications, including three internationally renowned books), and who has also been the Editor of an academic journal for many years, review this paper. Their suggested edits relating to English language and ‘flow’ have been included in this updated version. English is their first language.

This manuscript is a resubmission of an earlier submission. The following is a list of the peer review reports and author responses from that submission.

Round 1

Reviewer 1 Report

Comments and Suggestions for Authors

1. This paper explores the perceptions and attitudes of both peri-urban residents and growers/farmers, living and working within the peri-urban zone of ÅŒtautahi Christchurch. 

2. In order to explore the perceptions/attitudes of respondents, this paper design various types of questionnaires and analyze by different statistic methods. It is better if authors supplement a concise description for relation between questionnaire and statistic test.   

3. The arguments and discussion of findings are coherent.

4. The conclusions are thoroughly supported by the results presented in the article.

5. The word of “di” on Line 238 might be “did”.  

Reviewer 2 Report

Comments and Suggestions for Authors

Dear authors,

found the article to have an immense potential. It delves into the intricate dynamics between peri-urban farmers and residents – a pervasive global issue. By dissecting this phenomenon within the context of New Zealand, the article offers a noteworthy contribution to the ongoing discourse on this matter.

Nonetheless, the article falls short in its endeavour to contextualize the problem within a broader global framework. It could be strengthened by drawing parallels with findings from analogous research endeavours, identifying areas of knowledge deficiency, and underlining the research's significance transcending local boundaries.

The article is commendably structured and elucidates the research methods and diverse findings with precision. Yet, its tone leans more towards that of a research report rather than an engaging piece for a worldwide readership. The incorporation of more sophisticated visual aids beyond mere tables and graphs could elevate the article's appeal. Integrating advanced visual tools might augment its readability.

I do yearn for a richer visual portrayal, particularly capturing the tactile interface zones between the agricultural and residential domains. Supplementing the text with on-ground photographs could vividly encapsulate these pivotal junctures.

Lastly, the depth of analysis within the discussion segment leaves room for enhancement. While summarizing the findings, a more profound exploration of their ramifications would be valuable. The discussion could be elevated beyond mere exposition, delving into potential conflict resolution strategies. The concluding section, too, might extend its scope beyond commonplace generalities, embracing a more nuanced and pertinent recommendations or just directions for future research.

In summary, the article's latent potential is undeniable. With contextual enrichment, more engaging visuals, and a deeper analytical stance, it has the capacity to evolve from a well-crafted research report into a captivating piece with global resonance.

Reviewer 3 Report

Comments and Suggestions for Authors

Housing and Food Production: Resident and Grower Perceptions of Peri-urban Food-Production Landscapes

Based on the survey of peri-urban (three towns of Waikirikiri Selwyn District, Canterbury ÅŒtautahi Christchurch) residents and farm growers living in Aotearoa New Zealand, this research explores the perception and understanding of the food production nearby residents. Information was gathered using 2060 questionnaires of which 1423 were distributed to Rolleston, 430 to Lincoln, 165 and 207 to Darfield, which covers a similar percentage (approximately 5.5%) of the population in each town to reflect the relative population distribution by town and their perception toward fresh food production in nearby areas. The result outcomes have some merits that can help policymakers to design policies to improve the food chain supply system.  In conclusion, the authors have concluded that the perception of growing crops nearby has been positive and people are preferring fresh food grown in nearby areas rather than transported from long distances. This paper may contribute to sustainable development and improving food security while strengthening the food supply chain should cases like the Covid-19 pandemic prevails again. 

Comments on the Quality of English Language

This article needs thorough editing to a) make the flow smooth, b) shorten the length, and c) incorporate research findings in the abstract and conclusion.